



# Multilevel Monte Carlo methods for ensemble variational data assimilation

Mayeul Destouches[1,2], Paul Mycek[1], Selime Gürol[1], Anthony T. Weaver[1], Serge Gratton[3], and Ehouarn Simon[3]

[1]CERFACS/CECI CNRS UMR 5318, Toulouse, France
[2]Met Office, Exeter, United Kingdom
[3]INPT-IRIT, Toulouse, France

**Correspondence:** Mayeul Destouches (mayeul.destouches@umr-cnrm.fr)

**Abstract.**

Ensemble variational data assimilation relies on ensembles of forecasts to estimate the background error covariance matrix $\mathbf{B}$. The ensemble can be provided by an Ensemble of Data Assimilations (EDA), which runs independent perturbed data assimilation and forecast steps. The accuracy of the ensemble estimator of $\mathbf{B}$ is strongly limited by the small ensemble size that

is needed to keep the EDA computationally affordable We investigate here the potential of the multilevel Monte Carlo (MLMC) method, a type of multifidelity Monte Carlo method, to improve the accuracy of the standard Monte-Carlo estimator of $\mathbf{B}$ while keeping the computational cost of ensemble generation comparable. MLMC exploits the availability of a range of discretization grids, thus shifting part of the computational work from the original assimilation grid to coarser ones. MLMC differs from the mere averaging of statistical estimators, as it ensures that no bias from the coarse resolution grids is introduced in the estimation.

The implications for ensemble variational data assimilation systems based on EDAs are discussed. Numerical experiments with a quasi-geostrophic model demonstrate the potential of the approach, as MLMC yields more accurate background error covariances and reduced analysis error. The challenges involved in cycling a multilevel variational data assimilation system are identified and discussed.

## 1 Introduction

The covariance matrix of background errors plays a key role in variational data assimilation applications in meteorology

and oceanography (Bannister, 2008a, b). Historically, these covariance matrices were modelled with strong assumptions of homogeneity and isotropy, and with limited flow-dependence (e.g., by parameterizing covariances in terms of the background



state). Developments over the last twenty years have enabled operational numerical weather prediction (NWP) centres to incorporate information from an ensemble of forecasts into parameterized background error covariance models (e.g., Raynaud et al., 2009 and Bonavita et al., 2011 for global atmospheric data assimilation, Brousseau et al., 2012 for regional atmospheric
data assimilation, Chrust et al. (submitted) for global ocean data assimilation). Moving away from parameterized covariance models, Lorenc (2003) and Buehner (2005) showed how a sample-based covariance matrix estimated directly from an ensemble of forecasts could be used in variational data assimilation. This method, inspired by the ensemble Kalman filter, consists of regularizing the standard Monte Carlo (MC) estimator of the covariance matrix by explicitly damping covariances at long separation distances where the signal-to-noise ratio is small because of comparatively large sampling errors. This localized
ensemble representation of the background error covariance matrix is used by many NWP centres, including the Met Office, the UK's national weather service (Clayton et al., 2013), Environment and Climate Change Canada (ECCC, Buehner et al., 2015; Caron et al., 2015), the National Centers for Environmental Prediction (NCEP, Kleist and Ide, 2015) and Météo-France (Montmerle et al., 2018).

Ensemble generation strategies can be broadly divided in two classes, whether the ensemble members come from an en-
semble Kalman filter (or flavours thereof), as is done operationally at ECCC for instance (Buehner et al., 2015; Caron et al., 2022), or from an Ensemble of independent Data Assimilations (EDA). We will focus on the latter in this paper. EDAs are used operationally to initialize ensemble forecasts and to provide background error information at Météo-France (Pereira and Berre, 2006), ECMWF (Bonavita et al., 2015) and the Met Office (Inverarity et al., 2023). The advantages of an EDA include reduced maintenance cost (compared to maintaining two different schemes for the ensemble and deterministic systems), improved
simulation of the errors in the deterministic system, and its "embarrassingly parallel" nature.

Although ensemble estimates from an EDA can substantially improve the representation of background errors, this comes at the price of an increased computational cost, as a non-linear forecast and separate minimization (analysis) have to be performed for each ensemble member. Some research efforts have focused on reducing the cost of performing an ensemble of minimizations by using block-minimization methods (Mercier et al., 2019). A more typical way of reducing the combined cost
of the ensemble forecast generation and assimilation steps is to use a cheaper data assimilation algorithm and coarser resolution compared to a deterministic or control forecast (Michel and Brousseau, 2021). The "Mean-Pert" method (Lorenc et al., 2017) gives a theoretical basis to this approach, and has been used operationally at the Met Office since December 2019 (Inverarity et al., 2023). In this article, the focus is mostly on maintaining an affordable ensemble-generation cost.

In this paper, we explore a method to improve the estimation of background error covariances by increasing the ensemble
size without increasing the cost of generating the ensemble, in a way that is not so different from the dual-resolution strategy mentioned above. Instead of generating an ensemble on a single coarse grid, we generate a set of ensembles on grids with different levels of coarsening. The Multilevel Monte Carlo (MLMC) framework ensures that no bias is introduced in the resulting estimates. MLMC was first introduced by Heinrich (2000), and popularized by Giles (2008). A review is provided in Giles (2015). In practice, the MLMC estimator is built as a base MC estimator on a coarse grid, but is iteratively refined by
adding correction terms from the difference of pairs of MC estimators from finer and finer grids.





The idea of applying MLMC methods to data assimilation is not new. Several studies have examined MLMC in the context of particle filters (e.g., Gregory et al., 2016; Jasra et al., 2017) and of the Ensemble Kalman Filter(e.g., Hoel et al., 2016; Chernov et al., 2021). However, they often focus on the estimation of a scalar quantity, or on the asymptotic properties in the limit of continuous space or time discretizations. Here, we focus on the use of MLMC methods for variational data assimilation. In this regard, the natural extension of ensemble-variational (EnVar) systems is to use an MLMC estimator of the background error covariance matrix.

MLMC methods have also been used to estimate variances and covariances. Multilevel estimation of the variance field was analyzed by Bierig and Chernov (2015) (but used as early as 2010, see references therein). Mycek and de Lozzo (2019) proposed a multilevel estimator for scalar covariances, with a focus on the optimal allocation of members across grids. In the context of the EnKF, a multilevel estimator of the covariance matrix has been proposed by Hoel et al. (2016). Most interestingly, Maurais et al. (2023a, b) have recently proposed a multilevel estimator that can be used to ensure positive semi-definite covariance matrix estimates.

The main contribution of this paper is to explore the use of MLMC-like methods for variational data assimilation, which, to the best of our knowledge, has never been done before. In doing so, we impose constraints that are relevant for data assimilation in operational applications. For instance, we assume that the finest grid resolution of the data assimilation is fixed, so that only coarser grids can be added. As such, we do not delve into asymptotic considerations in the limit of infinitely many refined grids. As another example, we discard algorithms whose cost does not scale linearly or quasi-linearly with respect to the size of the state to estimate. For instance, our proposed algorithms do not involve explicit storage of the background error covariance matrix $\mathbf{B}$, nor performing eigen-decompositions of it or computing its eigen-decomposition. This concern for operational applications makes the spirit of this article similar to that of Beiser et al. (2024), who investigated the practical details of using MLMC with an EnKF in an oceanographic application.

The outline of the paper is as follows. We recall the properties of the standard single-level MC ensemble estimator of $\mathbf{B}$ in Sect. 2, before explaining in Sect. 3 how they extend to multilevel estimators. The setting for numerical experiments is presented in Sect. 4. Results are presented and discussed in Sect. 5 before the conclusion in Sect. 6.

## 2 The Monte Carlo estimator of B

This section recalls some statistical properties of the ensemble background error covariances classically used in ensemble variational data assimilation. The notations have been chosen to facilitate the extension to the multilevel setting in next section.

Let $\boldsymbol{\epsilon} : \Omega \to \mathbb{R}^p$ be a random vector representing stochastic model inputs, where $\Omega$ is the sample space. For instance, $\boldsymbol{\epsilon}$ could contain uncertainty information on initial conditions, boundary conditions and model parameters. Let $f : \mathbb{R}^p \to \mathbb{R}^n$ be a forecast model generating ensemble members $\boldsymbol{X} : \Omega \to \mathbb{R}^n$ from the stochastic inputs as $\boldsymbol{X} = f \circ \boldsymbol{\epsilon}$ where $\boldsymbol{X}$ follows the probability distribution of the background state, $n$ is the dimension of the state space, $p$ is the dimension of the space of uncertain input parameters, and $\circ$ is the composition operator. Our goal is to estimate the covariance matrix of $\boldsymbol{X}$, $\mathbf{B} := \mathrm{Cov}[\boldsymbol{X}, \boldsymbol{X}]$. We assume for the rest of the paper that all random vectors have finite fourth-order moments, and thus finite





second-order moments. Let $\mathcal{E} := (\boldsymbol{\epsilon}_1, \dots, \boldsymbol{\epsilon}_N)$ denote an ensemble of $N$ independent and identically distributed stochastic inputs, with $N \geq 2$. The standard $N$-member unbiased MC estimator of $\mathbf{B}$ can be built from the ensemble of $N$ forecast members associated to these inputs, $f(\mathcal{E}) := (f(\boldsymbol{\epsilon}_1), \dots, f(\boldsymbol{\epsilon}_N)) = (\boldsymbol{X}_1, \dots, \boldsymbol{X}_N)$.

$$\widehat{\mathbf{B}} = \frac{1}{N-1} \sum_{i=1}^{N} \left( \boldsymbol{X}_i - \overline{\boldsymbol{X}} \right) \left( \boldsymbol{X}_i - \overline{\boldsymbol{X}} \right)^{\mathsf{T}} \tag{1}$$

where $\overline{\boldsymbol{X}} := 1/N \sum_i \boldsymbol{X}_i$ denotes the ensemble mean.

## 2.1 Sampling noise

The MC estimator $\widehat{\mathbf{B}}$ is built from random vectors, so it is itself a random quantity. Each element $\widehat{B}_{ij}$ of the MC estimator of $\mathbf{B}$ is an estimator of $B_{ij}$. As for any statistical estimator, we can define its mean square error (MSE), which can be decomposed as a variance and a squared bias term:

$$\begin{aligned}\mathrm{MSE}\left[\widehat{B}_{ij}, B_{ij}\right] &:= \mathbb{E}\left[\left(\widehat{B}_{ij} - B_{ij}\right)^2\right] \\ &= \mathrm{Var}\left[\widehat{B}_{ij}\right] + \left(\mathbb{E}\left[\widehat{B}_{ij} - B_{ij}\right]\right)^2.\end{aligned}$$

This bias–variance decomposition is somewhat artificial here since the MC estimator of the covariance is unbiased. As a consequence, the MSE of the estimator reduces to its variance:

$$\mathrm{MSE}\left[\widehat{B}_{ij}, B_{ij}\right] = \mathrm{Var}\left[\widehat{B}_{ij}\right].$$

This variance is a measure of the amplitude of the sampling noise affecting the MC estimator and is not to be mistaken for the variance of $\mathbf{X}$, which is the diagonal of $\mathbf{B}$. Rather, it is the variance of the covariance estimator, which can be expressed as a function of ensemble size and the statistical moments of $\boldsymbol{X}$ (see Mycek and de Lozzo, 2019, Eq. 11):

$$\mathrm{Var}\left[\widehat{B}_{ij}\right] = \frac{1}{N} \mathbb{M}_{iijj}^{(4)} - \frac{N-2}{N(N-1)} B_{ij}^2 + \frac{1}{N(N-1)} B_{ii} B_{jj}, \tag{2}$$

where $\mathbb{M}^{(4)}$ is the tensor of fourth-order central moments of $\boldsymbol{X}$,

$$\mathbb{M}_{ijkl}^{(4)} := \mathbb{E}[(X_i - \mathbb{E}[X_i])(X_j - \mathbb{E}[X_j])(X_k - \mathbb{E}[X_k])(X_l - \mathbb{E}[X_l])].$$

Note that Eq. (2) is valid independently of the distribution of the ensemble members $\boldsymbol{X}$. We do not assume the members to be normally-distributed for instance. The MSE of the covariance estimator behaves asymptotically as $(\mathbb{M}_{iijj}^{(4)} - B_{ij}^2)/N$ as $N \to \infty$. The slow decay rate $\mathcal{O}\left(N^{-1/2}\right)$ of the *root* mean square error (RMSE) is a well-known property of MC estimators. In practice, this means that to reduce the sampling noise by half for instance, the ensemble size needs to be increased by a factor 4.



## 2.2 Total MSE and total variance

A natural choice to define the total MSE of the full covariance matrix estimator is to use the Frobenius norm $\|\mathbf{A}\|_{\mathrm{F}}^2 := \sum_i \sum_j A_{ij}^2$. With this choice of norm, the total MSE is just the sum of the matrix element MSEs, and a simple bias–variance decomposition follows:

$$
\begin{aligned}
\mathrm{MSE}\left[\widehat{\mathbf{B}}, \mathbf{B}\right] &:= \mathbb{E}\left[\left\|\widehat{\mathbf{B}} - \mathbf{B}\right\|_{\mathrm{F}}^2\right] \\
&= \mathbb{E}\left[\sum_{i=1}^n \sum_{j=1}^n \left(\widehat{B}_{ij} - B_{ij}\right)^2\right] \\
&= \sum_{i=1}^n \sum_{j=1}^n \mathrm{Var}\left[\widehat{B}_{ij}\right] + \left(\mathbb{E}\left[\widehat{B}_{ij} - B_{ij}\right]\right)^2 \\
&= \mathcal{V}\left[\widehat{\mathbf{B}}\right].
\end{aligned}
$$

The bias term vanishes in (3), and $\mathcal{V}$ denotes the total variance, defined as the sum of the element variances:

$$
\mathcal{V}\left[\widehat{\mathbf{B}}\right] := \sum_{i=1}^n \sum_{j=1}^n \mathrm{Var}\left[\widehat{B}_{ij}\right] \tag{3}
$$

$$
= \sum_{i=1}^n \sum_{j=1}^n \left[\frac{1}{N}\mathbb{M}_{iijj}^{(4)} - \frac{N-2}{N(N-1)}B_{ij}^2 + \frac{1}{N(N-1)}B_{ii}B_{jj}\right], \tag{4}
$$

where we have used expression (2).

## 2.3 Impact of localization

In practice, the MC covariance matrix estimate is never used as such. In data assimilation applications, the ensemble covariance matrix is regularized by localization (Hamill et al., 2001; Lorenc, 2003), i.e.,

$$
\widehat{\mathbf{B}}_{\mathrm{loc}} = \widehat{\mathbf{B}} \odot \mathbf{L}
$$

where the localization matrix $\mathbf{L} \in \mathbb{R}^{n \times n}$ is a correlation matrix and $\odot$ denotes the Schur (element-wise) product. Localization is designed to remove long-distance sampling noise, and in doing so increases the rank of the covariance matrix estimates. The localization matrix, which has elements between 0 and 1, introduces bias in the estimator $\widehat{\mathbf{B}}_{\mathrm{loc}}$. This additional bias may be compensated by an associated reduction of the variance. For any element $ij$, we have

$$
\mathbb{E}\left[\widehat{B}_{\mathrm{loc},ij} - B_{ij}\right] = \mathbb{E}\left[\widehat{B}_{ij}L_{ij} - B_{ij}\right] = \mathbb{E}\left[\widehat{B}_{ij}\right]L_{ij} - B_{ij} = B_{ij}(L_{ij} - 1),
$$

$$
\mathrm{Var}\left[\widehat{B}_{\mathrm{loc},ij}\right] = \mathrm{Var}\left[\widehat{B}_{ij}L_{ij}\right] = L_{ij}^2 \mathrm{Var}\left[\widehat{B}_{ij}\right].
$$

Ménétrier et al. (2015a, b) proposed a framework for optimal localization, based on minimizing the MSE of the localized estimator of $\widehat{\mathbf{B}}_{\mathrm{loc}}$.

In the next section, we focus on *non-localized* and *unbiased* estimators of $\mathbf{B}$. As a technical (though vital) tool, the interaction with localization will be detailed later, along with other technical considerations, in the numerical experiments of Sect. 5.



## 3   The multilevel B

In this section, we focus on using MLMC techniques (Heinrich, 2000; Giles, 2008, 2015) to build a multilevel $\mathbf{B}$ in order to reduce the total variance of the MC estimator without introducing any additional bias. The MC estimator can be seen as a specific case of a multilevel estimator with only one level. In Sect. 3.1 we first introduce the multilevel estimation of a covariance matrix. In Sect. 3.2 we derive the variance of this estimator and show how it can be minimized by an optimal allocation of the ensemble members on the different fidelity levels. Finally, a weighted extension of the classical multilevel estimator is introduced in Section 3.3.

### 3.1   Multilevel estimation of a covariance matrix

Contrary to simple MC estimators which are built from an ensemble of simulations of equal fidelity, MLMC estimators are built from a *set of ensembles* of simulations of different fidelity. In the multilevel approach, we introduce a hierarchy of models of increasing fidelities $f_1, \ldots, f_L$, with $L \geq 2$ and $f_L = f$. We assume that the models are ordered from the least (fidelity level $\ell = 1$) to the most accurate (fidelity level $\ell = L$), from the computationally cheapest ($\ell = 1$) to the most computationally expensive ($\ell = L$). These models have the same input and output spaces as $f$; i.e., $f_\ell : \mathbb{R}^p \to \mathbb{R}^n$. In practice, $f_1, \ldots, f_{L-1}$ are typically obtained by using intermediate coarser discretization grids. This implies that a prolongation operator is required to ensure that the output of each model is on the same fine grid in $\mathbb{R}^n$. The choice of interpolation operators will be discussed further in Sec. 4, and has been discussed in a similar context by Briant et al. (2023). The varying fidelity can originate from other sources than grid discretization, for instance from computations in simple or half precision, simplified dynamical cores or from surrogate models.

For each $1 \leq k \leq L$, we define a set of stochastic inputs with $N^{(k)}$ members in $\Omega \to \mathbb{R}^p$,

$$\mathcal{E}^{(k)} = \left( \boldsymbol{\epsilon}_1^{(k)}, \ldots, \boldsymbol{\epsilon}_{N^{(k)}}^{(k)} \right).$$

The total $\sum_{k=1}^{L} N^{(k)}$ stochastic inputs are all independent and identically distributed. Technically, each "stochastic input set" $\mathcal{E}^{(k)}$ could be given as input to any of the simulators $f_\ell$, thus yielding an output ensemble $\mathcal{X}_\ell^{(k)} := f_\ell(\mathcal{E}^{(k)})$. If the same stochastic input set $\mathcal{E}^{(k)}$ is given to several simulators, several output ensembles are produced, all of size $N^{(k)}$, all with their own fidelity level $\ell$, but all generated from the same stochastic inputs. As such, these ensembles would *not* be mutually independent, with one-to-one correlations between members across fidelity levels. We refer to such ensembles as "stochastically-coupled ensemble".

To build an MLMC estimator, we generate several groups of stochastically-coupled ensembles, one group for each of the $L$ stochastic input sets. A base group is defined using the lowest fidelity level only (group with one output ensemble only) and coupling groups are defined from pairs of successive fidelity levels (groups with two stochastically-coupled ensembles each):

Base group with one ensemble $\mathcal{X}_\ell^{(1)} = f_\ell\left(\mathcal{E}^{(1)}\right)$, $\ell = 1$;

Coupling group with stochastically-coupled ensemble pair $\left(\mathcal{X}_\ell^{(k)}, \mathcal{X}_{\ell'}^{(k)}\right)$, $\ell = k, \ell' = k-1$, $2 \leq k \leq L$.



The subscripts here refer to the fidelity level $\ell$, while the superscripts refer to the stochastic input set $k$. The coupled ensembles $\mathcal{X}_{k-1}^{(k)}$ and $\mathcal{X}_k^{(k)}$ are *not* independent as they use the same stochastic inputs, and thus are stochastically coupled. Conversely, ensembles with different superscripts are independent as they are generated from different stochastic input sets. As an example, the coupling structure of an MLMC with three fidelity levels is shown in Table 1.

| stochastic input set $k$ | fidelity level $\ell$ | ensemble sizes |
|---|---|---|
| 1 | 1 | $N^{(1)}$ |
| 2 | 1, 2 | $N^{(2)}$ each, $2N^{(2)}$ in total |
| 3 | 2, 3 | $N^{(3)}$ each, $2N^{(3)}$ in total |

**Table 1.** Example coupling structure for a 3-level MLMC estimator. Each row corresponds to a different coupling group.

Denoting by $\widehat{\mathbf{B}}_\ell^{(k)}$ the unbiased MC covariance estimator built from ensemble $\mathcal{X}_\ell^{(k)}$, the MLMC covariance estimator is built from a base estimator on the lowest-fidelity level, successively corrected by correction terms:

$$\widehat{\mathbf{B}}^{\mathrm{ML}} = \widehat{\mathbf{B}}_1^{(1)} + \sum_{k=2}^{L} \left( \widehat{\mathbf{B}}_k^{(k)} - \widehat{\mathbf{B}}_{k-1}^{(k)} \right). \tag{5}$$

There are a few important points to note about this estimator. First, building the ensembles requires $N^{(L)}$ integrations of the high-fidelity model $f_L = f$, and $N^{(\ell)} + N^{(\ell+1)}$ integrations of $f_\ell$, for $1 \le \ell \le L-1$. Compared to an MC estimator, this means that some computational budget must be moved from the high-fidelity level to the coarser ones to conserve the same computational cost. In practice, the low-fidelity models are much cheaper to run, which allows them to be run with large ensembles even with a fraction of the total computational resources. Second, This estimator is unbiased, as the expectations are not dependent on the stochastic input index and cancel each other out in a telescopic fashion:

$$\mathbb{E}\left[\widehat{\mathbf{B}}^{\mathrm{ML}}\right] = \mathbb{E}\left[\widehat{\mathbf{B}}_1^{(1)}\right] + \sum_{k=2}^{L} \left( \mathbb{E}\left[\widehat{\mathbf{B}}_k^{(k)}\right] - \mathbb{E}\left[\widehat{\mathbf{B}}_{k-1}^{(k)}\right] \right)$$

$$= \mathbf{B}_1 + \sum_{k=2}^{L} (\mathbf{B}_k - \mathbf{B}_{k-1}) \qquad (\text{where } \mathbf{B}_\ell := \mathrm{Cov}[f_\ell(\boldsymbol{\epsilon}), f_\ell(\boldsymbol{\epsilon})])$$

$$= \mathbf{B}_L = \mathbf{B}.$$

where we used the fact that although $\widehat{\mathbf{B}}_{k-1}^{(k-1)} \ne \widehat{\mathbf{B}}_{k-1}^{(k)}$, they have the same expectation $\mathbf{B}_{k-1}$. Third, it is not obvious under which conditions this multilevel estimator is more accurate than the standard MC one. This is discussed in Sect. 3.2.

Finally, multilevel estimators are not range-preserving. If random samples lie in a given interval, an MC estimate of their mean will also lie in this interval, but an MLMC estimate may lie outside the interval. This concept generalizes to covariance matrix estimators, with multilevel estimates being symmetric matrices with no guarantee of positive semi-definiteness. This is discussed further in Sect. 5.



## 3.2 Error reduction and optimal ensemble sizes

There are various ways to split a given computational budget into the different coupling groups. How the computational budget
is allocated determines the respective ensemble sizes $N^{(k)}$, which determine in turn the MSE of the multilevel covariance
estimator. We will explain in this section how the member allocation (i.e., choice of the ensemble sizes) can be optimized
to minimize the statistical error (sampling noise) of the covariance matrix estimator. The approach we follow here is similar
to what can be found in the literature, for instance in Mycek and de Lozzo (2019) for the multilevel estimation of a scalar
covariance. The results in this section, taken from Destouches et al. (2023), differ from previously published results in two
aspects: a) we extend the problem from the *scalar* multilevel covariance estimator to the multilevel covariance *matrix* estimator,
in a setting where the finest level $L$ is fixed, and b) we ensure that the problem is kept numerically feasible for high-dimensional
systems.

First, let us compute the MSE of the covariance matrix estimator. Since the multilevel covariance estimator is unbiased, its
total MSE is equal to its total variance. The mutual independence of correction terms allows us to write the total variance as

$$\mathcal{V}\left[\widehat{\mathbf{B}}^{\mathrm{ML}}\right] = \mathcal{V}\left[\widehat{\mathbf{B}}_1^{(1)}\right] + \sum_{k=2}^{L} \mathcal{V}\left[\widehat{\mathbf{B}}_k^{(k)} - \widehat{\mathbf{B}}_{k-1}^{(k)}\right]. \tag{6}$$

As shown by Mycek and de Lozzo (2019, Equation 2.31) for a scalar covariance estimator, the scalar variance of each scalar
multilevel covariance estimator $\widehat{B}_{ij}^{\mathrm{ML}}$ can be bounded by a sum over the coupling groups $k$ of terms inversely proportional to
$N^{(k)} - 1$. By summing these terms over all covariances $\widehat{B}_{ij}^{\mathrm{ML}}$ in $\widehat{\mathbf{B}}^{\mathrm{ML}}$, we see the same result holds for the *total* variance of the
covariance matrix estimator:

$$\mathcal{V}\left[\widehat{\mathbf{B}}^{\mathrm{ML}}\right] \leq \sum_{k=1}^{L} \frac{\mathcal{V}^{(k)}}{N^{(k)} - 1}, \tag{7}$$

where $\mathcal{V}^{(k)}$ are constants independent of the ensemble sizes. These constants can be expressed as functions of central moments
of $f_\ell(\boldsymbol{\epsilon})$. Introducing $\boldsymbol{X}_\ell := f_\ell(\boldsymbol{\epsilon})$ and $X_{\ell,i}$ its $i$-th element, and defining $X_{0,i} = 0$ by convention for all $i$, we have, for $k \geq 1$,

$$\mathcal{V}^{(k)} := \sum_{1 \leq i,j \leq n} \frac{1}{2} \left( \sqrt{\mathbb{M}^4[X_{k,i} - X_{k-1,i}] \, \mathbb{M}^4[X_{k,j} + X_{k-1,j}]} + \sqrt{\mathbb{M}^4[X_{k,j} - X_{k-1,j}] \, \mathbb{M}^4[X_{k,i} + X_{k-1,i}]} \right),$$

where $\mathbb{M}^4$ is the fourth-order central statistical moment: $\mathbb{M}^4[u,v,x,y] := \mathbb{E}[(u - \mathbb{E}[u])(v - \mathbb{E}[v])(x - \mathbb{E}[x])(y - \mathbb{E}[y])]$ for
scalar, real-valued random variables $u$, $v$, $x$ and $y$, and $\mathbb{M}^4[u] := \mathbb{M}^4[u,u,u,u]$.

Although we can derive an exact expression of the total variance (see below), Eq. (7) is helpful to understand why the
multilevel approach can be effective. Intuitively, the upper bound will be small if either the ensemble sizes $N^{(k)}$ are large
or the constants $\mathcal{V}^{(k)}$ are small. The first condition can be met for small $k$ if the low-fidelity simulations are substantially
cheaper to run than the high-fidelity ones, thus allowing large ensemble sizes for low-fidelity ensembles. For higher fidelity
ensembles, the ensemble sizes are necessarily smaller, due to higher computational cost. This can be compensated by smaller
$\mathcal{V}^{(k)}$ constants, which can be related to small fourth-order moments of the correction terms, and so to strong correlations
between stochastically-coupled simulations. However, building computationally cheaper low-fidelity models that are strongly
correlated with the highest fidelity model can be technically challenging.





The *exact* total variance can be computed by expanding Eq. (6):

$$\mathcal{V}\left[\widehat{\mathbf{B}}^{\mathrm{ML}}\right] = \sum_{1 \le i,j \le n} \left\{ \mathbb{V}\left[\widehat{B}_{1,ij}^{(1)}\right] + \sum_{k=2}^{L} \left( \mathbb{V}\left[\widehat{B}_{k,ij}^{(k)}\right] + \mathbb{V}\left[\widehat{B}_{k-1,ij}^{(k)}\right] - 2\,\mathrm{Cov}\left[\widehat{B}_{k,ij}^{(k)}, \widehat{B}_{k-1,ij}^{(k)}\right] \right) \right\}. \tag{8}$$

All the terms in this expression can be expressed as functions of the ensemble sizes $N^{(k)}$ and central moments of $\boldsymbol{X}_\ell$. We can derive the following expression for the covariance of scalar covariance estimators (Ménétrier et al., 2015a, their Eq. 9):

$$\mathrm{Cov}\left[\widehat{B}_{\ell,ij}^{(k)}, \widehat{B}_{\ell',ij}^{(k)}\right] = \frac{\mathbb{M}^4[X_{\ell,i}, X_{\ell',i}, X_{\ell,j}, X_{\ell',j}] - \mathrm{Cov}[X_{\ell,i}, X_{\ell,j}]\,\mathrm{Cov}[X_{\ell',i}, X_{\ell',j}]}{N^{(k)}}$$
$$+ \frac{\mathrm{Cov}[X_{\ell,i}, X_{\ell',j}]\,\mathrm{Cov}[X_{\ell,j}, X_{\ell',i}] + \mathrm{Cov}[X_{\ell,i}, X_{\ell',i}]\,\mathrm{Cov}[X_{\ell,j}, X_{\ell',j}]}{N^{(k)}(N^{(k)} - 1)}. \tag{9}$$

For the case $\ell = \ell'$, this simplifies to an expression for the variance:

$$\mathrm{Var}\left[\widehat{B}_{\ell,ij}^{(k)}\right] = \frac{\mathbb{M}^4\left[X_{\ell,i}, X_{\ell,i}, X_{\ell,j}, X_{\ell,j}\right] - \mathrm{Cov}\left[X_{\ell,i}, X_{\ell,j}\right]^2}{N^{(k)}} + \frac{\mathrm{Cov}\left[X_{\ell,i}, X_{\ell,j}\right]^2 + \mathbb{V}\left[X_{\ell,i}\right]\mathbb{V}\left[X_{\ell,j}\right]}{N^{(k)}(N^{(k)} - 1)}. \tag{10}$$

Gathering Eq. (8)–(10) gives a theoretical expression of the total variance of the multilevel estimator as

$$\mathcal{V}\left[\widehat{\mathbf{B}}^{\mathrm{ML}}\right] = \sum_{k=1}^{L} \frac{a^{(k)}}{N^{(k)}} + \frac{b^{(k)}}{N^{(k)}(N^{(k)} - 1)} \tag{11}$$

for some deterministic scalar parameters $a^{(k)}$ and $b^{(k)}$, $1 \le k \le L$, which are independent of the ensemble sizes $N^{(k)}$ and given by

$$a^{(k)} = \sum_{1 \le i,j \le n} \Big( \mathbb{M}^4[X_{k,i}, X_{k,i}, X_{k,j}, X_{k,j}] - \mathrm{Cov}[X_{k,i}, X_{k,j}]^2 + \mathbb{M}^4[X_{k-1,i}, X_{k-1,i}, X_{k-1,j}, X_{k-1,j}]$$
$$- \mathrm{Cov}[X_{k-1,i}, X_{k-1,j}]^2 - 2\mathbb{M}^4[X_{k,i}, X_{k-1,i}, X_{k,j}, X_{k-1,j}] + 2\,\mathrm{Cov}[X_{k,i}, X_{k,j}]\,\mathrm{Cov}[X_{k-1,i}, X_{k-1,j}] \Big), \tag{12}$$

$$b^{(k)} = \sum_{1 \le i,j \le n} \Big( \mathrm{Cov}[X_{k,i}, X_{k,j}]^2 + \mathbb{V}[X_{k,i}]\mathbb{V}[X_{k,j}] + \mathrm{Cov}[X_{k-1,i}, X_{k-1,j}]^2 + \mathbb{V}[X_{k-1,i}]\mathbb{V}[X_{k-1,j}]$$
$$- 2\,\mathrm{Cov}[X_{k,i}, X_{k-1,j}]\,\mathrm{Cov}[X_{k,j}, X_{k-1,i}] - 2\,\mathrm{Cov}[X_{k,i}, X_{k-1,i}]\,\mathrm{Cov}[X_{k,j}, X_{k-1,j}] \Big). \tag{13}$$

where we define by convention $X_{0,i} = 0$ for all $i$. These expressions involve double sums over the $n$ elements of $\boldsymbol{X}_\ell$, i.e., over the space dimension. For operational systems of typical size $\mathcal{O}(10^6) - \mathcal{O}(10^9)$, these summations involve $\mathcal{O}(10^{12}) - \mathcal{O}(10^{18})$ elements. Evaluating these summations is computationally prohibitive. Fortunately, when estimating the statistical moments in Eq. (9)–(10) using ensemble estimates, sums can be rearranged to replace the double sums over space by a single sum over space, at the expense of a double sum over the ensemble members. More details on this are given in Destouches et al. (2023, Appendix B). Note that the natural ensemble estimators of $a^{(k)}$ and $b^{(k)}$ are biased. These "natural estimators" are obtained by replacing the central moments in (12) and (13) with the associated sample central moments about the average. The resulting estimators could be made unbiased by extending the approach of Shivanand (2023), but we have preferred using the biased versions here for the sake of simplicity.





Obtaining accurate ensemble estimates of fourth-order moments in Eq. (9)–(10) could also be difficult with small ensemble sizes. We do not expect this to be a problem in practice, however, as the sampling noise in fourth-order estimates is attenuated by the global space averaging on indexes $i$ and $j$.

Given that the total variance of the multilevel covariance matrix estimator is explicitly derived in Eq. (11), the optimal member allocation for a given computational budget $\eta$ can be determined by minimizing the total variance under a cost constraint:

$$255 \quad \min_{N^{(k)}, 1 \leq k \leq L} \sum_{k=1}^{L} \frac{a^{(k)}}{N^{(k)}} + \frac{b^{(k)}}{N^{(k)}(N^{(k)} - 1)} \quad \text{subject to} \quad \sum_{k=1}^{L} (\mathcal{C}_\ell + \mathcal{C}_{\ell-1}) N^{(k)} = \eta \text{ and } N^{(k)} \geq 2 \ (1 \leq k \leq N) \tag{14}$$

where $\mathcal{C}_\ell$ is the computational cost of generating one member on fidelity level $\ell$, and $\mathcal{C}_0 = 0$ by convention. This is a low-dimensional optimization problem, with 2 to 5 unknowns in practice, which can be solved numerically. The solution of the problem gives ensemble sizes in floating points, which can then be rounded to the closest integers, for instance. Only once this problem has been solved can the optimum variance be compared to the variance of the same cost MC estimator, obtained by
260 choosing $N$ as $\eta/\mathcal{C}_L$ in Eq. (4).

### 3.3 Weighted multilevel estimation

Schaden and Ullmann (2020) present an important generalization of the MLMC framework. In their paper, the authors introduce the multilevel best linear unbiased estimator (MLBLUE), a unifying framework that generalizes and outperforms the MLMC estimator as presented here. Although their work focuses on the estimation of a scalar expectation, it can be extended
to the estimation of a covariance matrix, as shown by Destouches et al. (2023).

In an MLMC framework, the fidelity levels are ordered from low to high fidelity, and successive fidelity levels are paired into a common coupling group receiving the same stochastic inputs. Each coupling group forms a correction term in Eq. (5). The MLBLUE does not make such assumptions: the fidelity levels do not need to be hierarchized, and a coupling group can include any number of levels. This is relevant, for example, when using non-hierarchized surrogate models (e.g., El Amri et al., 2023).
As a consequence, the corrections terms are no longer restricted to differences of MC estimators, but can be any weighted average of them, with coupling weights differing from the MLMC weights $(1, -1)$.

In the present paper, we do have hierarchized fidelity levels. As such, we choose to consider the MLBLUE only in the context of simpler MLMC-like coupling structures. Two main reasons support this choice.

1. For a given set of $L$ fidelity levels, the MLMC structure ensures that the number $(2L - 1)$ of different ensembles is
275 limited, while it could have been larger for more complex structures. In a context where the total computational budget is expected to be small (typically around 10 fine-grid simulations), keeping the number of ensembles small ensures each ensemble can be populated with a reasonable number of members;

2. Whichever coupling structure is chosen, it has to be kept unchanged for all data assimilation cycles. Since the optimal MLBLUE coupling structure is highly dependent on the problem under consideration, a complex coupling structure
that is optimal for one cycle may not be optimal for the other cycles. This motivates the choice of a simple and robust coupling structure.





Within this MLMC-like structure, the MLBLUE still provides an improvement over MLMC due to the introduction of coupling weights. This yields a weighted MLMC, as already proposed by Šukys et al. (2017). For a covariance matrix, a simplified weighted estimator with scalar weights reads

$$\widehat{\mathbf{B}}^{\text{wML}} = \beta_1^{(1)}\widehat{\mathbf{B}}_1^{(1)} + \sum_{k=2}^{L}\left(\beta_k^{(k)}\widehat{\mathbf{B}}_k^{(k)} + \beta_{k-1}^{(k)}\widehat{\mathbf{B}}_{k-1}^{(k)}\right) \tag{15}$$

where the $\beta_\ell^{(k)}$ are scalar weights, with $\beta_L^{(L)} = 1$ and $\beta_k^{(k+1)} = -\beta_k^{(k)}$ ($1 \leq k \leq L-1$) to ensure the estimator is unbiased. The variance of this estimator now depends on both the ensemble sizes $N^{(k)}$ and the weights $\beta_\ell^{(k)}$. However, the optimal weights can be derived for arbitrary ensemble sizes, so that the variance minimization problem can be expressed in terms of ensemble sizes only (Schaden and Ullmann, 2020).

For the simple weighted multilevel covariance matrix estimator proposed above, the ensemble member allocation problem is similar to the MLMC one (Destouches et al., 2023, translating their Eq. 124 to our notations):

$$\min_{N^{(k)}, 1 \leq k \leq N} \sum_{k=1}^{L}\begin{pmatrix} \beta_{k-1}^{(k)*} & \beta_k^{(k)*} \end{pmatrix} \mathbf{C}^{(k)}\begin{pmatrix} \beta_{k-1}^{(k)*} \\ \beta_k^{(k)*} \end{pmatrix} \quad \text{subject to} \quad \sum_{k=1}^{L}(\mathcal{C}_\ell + \mathcal{C}_{\ell-1})N^{(k)} = \eta \text{ and } N^{(k)} \geq 2 \, (1 \leq k \leq N) \tag{16}$$

where

- $\mathbf{C}^{(k)}$ is the matrix of spatially-averaged covariances between MC covariance estimators at successive fidelity levels,

$$\mathbf{C}^{(k)} = \frac{1}{n^2}\sum_{1 \leq i,j \leq n}\begin{pmatrix} \text{Var}\left[\widehat{B}_{k-1,ij}\right] & \text{Cov}\left[\widehat{B}_{k-1,ij}, \widehat{B}_{k,ij}\right] \\ \text{Cov}\left[\widehat{B}_{k-1,ij}, \widehat{B}_{k,ij}\right] & \text{Var}\left[\widehat{B}_{k,ij}\right] \end{pmatrix}, \, 1 \leq k \leq L;$$

- The quantities indexed by $k = 0$ are zero by convention;

- $\beta_\ell^{(k)*}$ are the optimal weights associated with the ensemble sizes $N^{(k)}$. Their expression is given in appendix A.

In the minimization problem (16), the ensemble sizes $N^{(k)}$ appear explicitly in the constraint, but they are only implicitly present in the optimal weights $\beta_\ell^{(k)*}$ (cf. App. A), and in the inter-level covariances (cf. Eq. 9). The summed inter-level covariance matrices $\mathbf{C}^{(k)}$ have to be estimated in a pre-processing step. This estimation can be made with a cost that is linear in $n$ (cf. Appendix B of Destouches et al., 2023). Since there is almost no cost overhead in using weighted MLMC over MLMC, the weighted MLMC (wMLMC hereafter) should always be preferred in practice.



# 4 Experimental setting

## 4.1 Presentation of the idealized model

The performance of the covariance estimators is tested with a two-dimensional two-layer quasi-geostrophic channel model. This simplified representation of the mid-latitude dynamics of the atmosphere is based on the conservation of potential vorticity in two well-mixed layers with uniform potential temperature on a $\beta$-plane (Fandry and Leslie, 1984, their section 2a). In this setting, the atmosphere can be represented either by the stream function field $\phi(x, y, z, t)$ or by the potential vorticity field $q(x, y, z, t)$.

Similar implementations as the one chosen in this article have been used at ECMWF by Fisher and Gürol (2017), Laloyaux et al. (2020), Farchi et al. (2021) and Farchi et al. (2023), to explore the impact of 4DVar research before trialling it in operational systems. In this article, we use the model implemented in the OOPS (Object-Oriented Prediction System) repository of the Joint Center for Satellite Data Assimilation (JCSDA). It is derived from the implementation presented in Fisher and Gürol (2017), but uses dimensional variables instead of non-dimensional variables. The only other differences are the domain

dimensions, grid sizes and time step which have been adapted to accommodate easier addition of coarser grids.

The chosen domain is rectangular, with periodic boundary conditions on the East-West direction. The zonal dimension is chosen to be the latitude circle length at 43 degrees North ($L_x = 29,277$ km), and the meridional dimension is chosen to ensure a 1:3 ratio ($L_y = 9,759$ km). The domain is discretized on a $n_y \times n_x = 80 \times 240$ grid, with data points defined at the nodes of the grid. Vertically, the two model levels have depths of 4 km (bottom level) and 6 km (top level), as in Fisher and Gürol (2017).

The northward winds are assumed to be zero on the North and South boundaries. This implies that the stream function has uniform values on these boundaries. The Dirichlet boundary conditions on stream function are chosen to impose an average eastward wind speed of 10 m/s in the lower level and 40 m/s in the upper level. A potential vorticity forcing term is added to the bottom layer equations, corresponding to a Gaussian heating or orography source of e-folding radius 1000 km and of amplitude $5 \times 10^{-5}$ s$^{-1}$, centred on a grid point located at one quarter of the domain in the eastward direction and three quarters of the

domain in the northward direction.

The time integration is based on a semi-Lagrangian upstream scheme to advect potential vorticity. Each time step includes a conversion from stream function to potential vorticity and horizontal winds, an advection of potential vorticity by the winds, and a conversion of the advected potential vorticity into stream function. This last step requires solving a Poisson equation. Preliminary numerical experiments showed that the accuracy of the numerical integration was not improving substantially for

time steps smaller than 5 minutes. This 5 minute time stepping has thus been retained.

The code to reproduce these results is publicly available, as stated in the "code availability" statement at the end of this paper.

## 4.2 Reference experiments

The experiment presented here are all based on pure 3DEnVar (three-dimensional ensemble variational) analysis schemes,

i.e., 3DVar schemes that rely on a purely ensemble-based background error covariance matrix, with no static (parametric)





covariance term. Performance is measured from the analysis error with respect to a truth run. We focus on a single analysis in this article and do not investigate the impact on a cycled forecast-analysis system.

Hereafter, all computations (performance metrics, minimizations, field perturbations, interpolations) use the stream function representation of the model, consistently with the aforementioned ECMWF studies. Since stream function is a horizontally
integrated quantity compared to potential vorticity, it effectively transforms positioning errors into amplitude errors, and thus is better suited to Gaussian-based variational assimilation and to root-mean-square performance metrics. The stream function also offers a representation of the system state with larger length-scales than potential vorticity, and is therefore better suited to spatial interpolation.

The truth run is initialized by first integrating the model over 60 days to reach a permanent regime. From this initial 60-day
forecast, the truth run is integrated forward for an additional 12 hours, from time $t = 0$ to time $t = 12$ hours. A background state is generated by adding a perturbation field to the initial true state at $t = 0$. This perturbation field is generated from a Gaussian covariance model with length-scales of 1000 km horizontally and 6 km vertically, and with standard deviation $6 \times 10^6$ m$^2$s$^{-1}$. For consistency with the unperturbed North and South boundary conditions, the standard deviations of the covariance model decay linearly to zero within 300 km of the North and South boundaries. This affects the two rows of grid points in the vicinity
of each boundary. A simplified ensemble generation scheme is devised by adding similar perturbations to the already-perturbed background state at $t = 0$, and integrating forward in time for 12 hours. The resulting ensemble is Gaussian-distributed at $t = 0$, and weakly non-Gaussian at $t = 12$ h due to the non-linear model integrations. The assimilation step is performed at $t = 12$ h.

In addition to the multilevel estimators (MLMC and wMLMC), two reference **B** matrices are computed at the analysis time: an MC estimator with 20 members (referred to as MC) and a reference MC estimator with $10^4$ members (referred to as 10k
MC). The 20-member MC estimator is our baseline, with 20 fine-grid members being the target budget we can afford. The 10k MC estimator will be used as an estimation of the true covariances.

Details on the analysis settings, including on covariance localization, will be given in Sect. 5.

### 4.3 Building a hierarchy of low-fidelity models

In order to improve upon the MC estimation of background error covariances, we need to introduce a hierarchy of low-fidelity
models. Here, this hierarchy is provided by coarsening the horizontal model grid and time resolution over the 12 h integration period. Details on the hierarchy of discretizations are given in Table 2. A spatial coarsening ratio of 2 is used between each fidelity level, so that grid points on coarse grids are a subset of the grid points on finer grids. For fidelity level $\ell$, $1 \leq \ell \leq 4$ the grid points are positioned at

$$
\begin{cases}
x_i(\ell) = i \cdot h_x(\ell),\ 0 \leq i \leq n_x(\ell) - 1 \\
y_i(\ell) = i \cdot h_y(\ell),\ 1 \leq i \leq n_y(\ell) - 1
\end{cases}
\tag{17}
$$

where the grid cell sizes are defined as $h_x(\ell) = L_x/n_x(\ell)$ and $h_y(\ell) = L_y/n_y(\ell)$. In practice, there is a 1:3 ratio between $n_x$ and $n_y$, and between $L_x$ and $L_y$, so that $h_x = h_y$ for all fidelity levels. Note that Equations (17) are also valid for $\ell = 4$, i.e., for the fine discretization introduced in Sect. 4.1.



For each low-resolution grid, independent experiments have been run to choose an integration time step $\Delta t(\ell)$. This time step has been chosen as the largest time step that kept the time-discretization error significantly smaller than the space-discretization
error. The time step lengths $\Delta t(\ell)$ are given in Table 2, alongside the discretization grids.

**Table 2.** Space and time discretizations for the low-fidelity models. Fidelity level 1 refers to the lowest fidelity (coarsest grid), while fidelity level 4 refers to the highest fidelity (original model grid). The degrees of freedom is the dimension of the state space, i.e., the number of free grid parameters $n_x(\ell) \times (n_y(\ell) - 1) \times 2$. The normalized integration cost is given as a fraction of the cost of a fine-grid run.

| fidelity level $\ell$ | $n_x(\ell)$ | $n_y(\ell)$ | $\Delta t(\ell)$ | degrees of freedom | normalized integration cost |
|---|---|---|---|---|---|
| 1 | 30 | 10 | 40 min | 540 | $1.78 \times 10^{-3}$ |
| 2 | 60 | 20 | 20 min | 2,280 | 0.0150 |
| 3 | 120 | 40 | 10 min | 9,360 | 0.123 |
| 4 | 240 | 80 | 5 min | 37,920 | 1 |

As the respective sizes of the inputs and outputs of the models must be identical across fidelity levels, bicubic interpolation operators are used to define restriction and prolongation between the fine grid (fidelity level 4) and coarser grids. Since the stencil of this interpolation is of size 4 by 4, an additional row of data is needed at the North and South of the domain to perform the interpolation. These rows are added before the bicubic interpolation by linear extrapolation. Using a higher-order
interpolation than the simple bilinear interpolation ensures the interpolated fields are smoothed, thus damping (at least partially) spurious high frequencies that can contaminate the analysis as illustrated by Briant et al. (2023).

To compare the cost of running a 20-member ensemble on the fine grid with a multilevel ensemble on all grids, we need to quantify the cost of generating an ensemble member on each grid. In practice, the wall-clock time of running a quasi-geostrophic forecast on the two coarsest grids is almost identical. This shows that the computational cost is not dominated
by the number of grid points at these coarse (and unrealistic) resolutions. To alleviate this, we chose the more realistic and favourable cost model where the computational cost scales as the number of grid points times the number of time steps over the integration period. This theoretical computational cost of running a 12 h forecast on each grid is given in Table 2.

In this experimental setting, the stochastic coupling exploited by the multilevel approach is based on ensembles on two fidelity levels using the same set of initial conditions. The coupling is strongest before the model integration (almost perfect
correlation, up to interpolation errors), and decays with increasing forecast lead time. An illustration of a coupled ensemble across all 4 fidelity levels is shown in Fig. 1. The vorticity fields are shown here, as they make small details more visible to the eye. All ensemble members are defined on the fine grid, even if they have been computed on a coarser grid, since the low-fidelity models include the restriction and prolongation operators. For each fidelity level, a pair of ensemble members is shown. The difference between the two ensemble members on the fine grid ($\ell = 4$) gives an estimate of the ensemble spread.
The associated coupled members on the next coarsest grid ($\ell = 3$) show some correlations with the fine grid simulations, as evidenced for instance by the position of the vortices. Less signal is preserved for the next coarsest grids. Note that when building a multilevel estimator, coupled simulations are built only for adjacent fidelity levels. Coupled members across 4



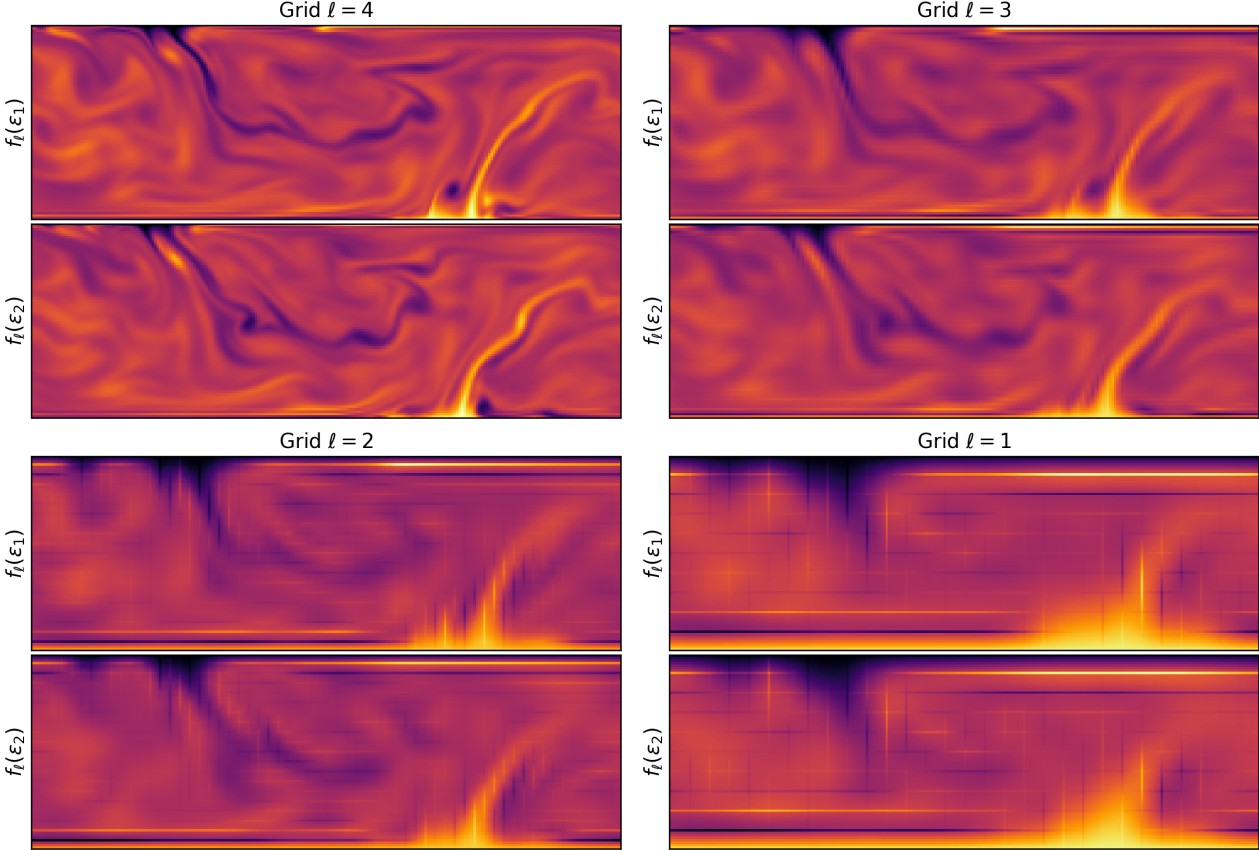

**Figure 1.** Two ensemble members after a 12h forecast, with associated coupled members at coarser resolutions. The vorticity field of the bottom model level is shown here.

different levels would never be used, except to compute statistics as an offline pre-processing step. In this 4-level setting, only the correlations between levels 4 and 3, 3 and 2, or 2 and 1 are of importance.

The vertical and horizontal lines visible on grids $\ell = 1$ and $\ell = 2$ reveal discontinuities in the potential vorticity fields, i.e., in the second-order spatial derivatives of the stream function field. This is consistent with the bicubic interpolation not guaranteeing continuity of spatial derivatives.

## 5    Numerical results

This section is divided into four parts. In a pre-processing stage, we first use a multilevel ensemble coupled across all levels to
estimate the optimal ensemble sizes in each coupling group. Optimality is defined here in terms of accuracy of the covariance





estimator. Building on this optimal allocation, we then verify that the associated multilevel covariance estimator is indeed more accurate than its MC counterpart. Before introducing this multilevel covariance estimator into a variational analysis scheme, we discuss the lack of positive semi-definiteness of the covariance estimates, and how this could be circumvented. We then assess the impact of using a multilevel covariance matrix $\mathbf{B}$ on one analysis.

## 5.1 Optimal member allocation

The optimal member allocation is computed from a set of ensembles stochastically coupled across all levels. We used 100 coupled members for this estimation. This value is not unrealistic, as this can typically be computed offline.

From this coupled ensemble, we estimate summed inter-level covariances of MC covariance estimators, as described in Appendix B of Destouches et al. (2023). Using these statistics, we can evaluate the total variance of a multilevel covariance estimator, and minimize it under a cost constraint (Eq. 14 for MLMC, Eq. 16 for wMLMC). The sampling noise of this estimation is much smaller than the sampling noises we try to minimize, due to the spatial averaging. Here, we impose the ensemble generation cost to be equal to that of a 20-member fine-grid MC estimator. We first solve the problem on real numbers using sequential least square programming, and then round the solution to the nearest integer allocations. As this may result in a computational budget slightly different than the target cost of 20 ensemble members, we fine-tune the allocation by removing or adding members to ensure we stay below the target budget while having as many ensemble members as possible on each coupling group.

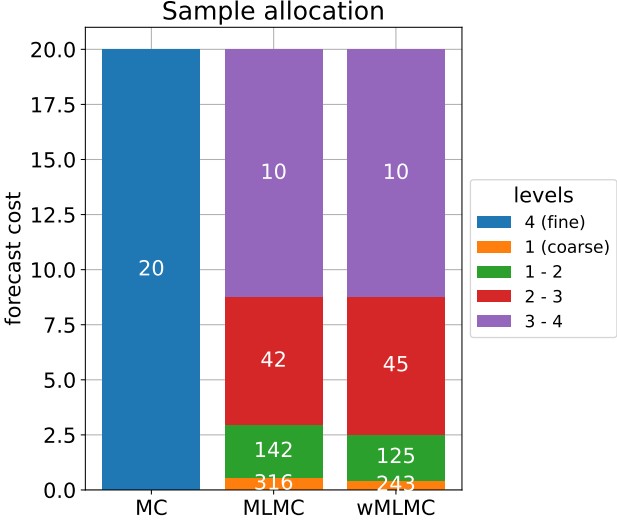

**Figure 2.** Allocation of members in different ensemble groups for different ensemble estimators: MC, MLMC and wMLMC. The height of each bar is proportional to the computational cost of generating the corresponding coupled ensemble. The number of members in each coupled ensemble is reported on each bar.




The optimal ensemble sizes and the associated generation costs are shown in Fig. 2. The member allocation is similar for MLMC and wMLMC. More than half of the computational resources are dedicated to running a 10-member coupled ensemble on the finest coupling group: 10 members on the fine grid $\ell = 4$ and 10 coupled members on grid $\ell = 3$. The ensemble sizes

increase as the associated generation cost decreases, until the coarser coupling group, where hundreds of ensemble members can be run for the cost of a single fine-grid member.

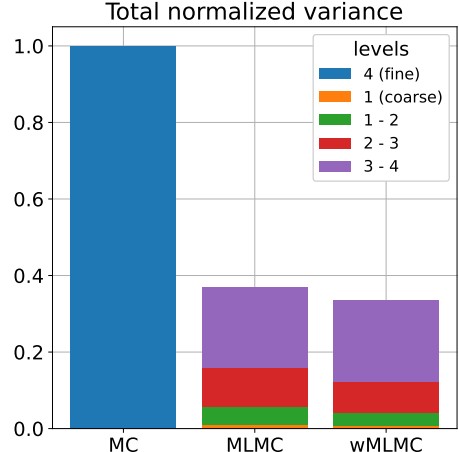

**Figure 3.** Theoretical estimate of the total variance of the covariance matrix estimators introduced in Fig. 2, normalized by the variance of the MC estimator. The total variances are estimated from Eq. (4) for MC, Eq. (11) for MLMC, and the function to minimize in Eq. (16) for wMLMC.

The total variance of the covariance estimator is shown in Fig. 3. The total variance is approximately three times smaller for the wMLMC estimator (the exact ratio being 0.337). To reach such a low variance, the single-level MC estimator would have required 60 ensemble members instead of 20.

The weights of the wMLMC associated with this optimal member allocation are, from the coarsest to the finest level: $\beta_1^{(1)} = -\beta_1^{(2)} = 0.70$, $\beta_2^{(2)} = -\beta_2^{(3)} = 0.72$, $\beta_3^{(3)} = -\beta_3^{(4)} = 0.81$ and $\beta_4^{(4)} = 1$. The benefit of wMLMC over MLMC is only marginal here, increasing the gain from 63% to 66% variance reduction. This could have been different under less favorable circumstances. For instance, doubling the forecast integration time to 24 hours instead of 12 hours would weaken the stochastic coupling and produce an optimal variance reduction of 38% for MLMC and 51% for wMLMC, i.e., a 13-point gain by adding

optimal weights to each level. Only wMLMC is considered in the rest of this section.

Note that there was no guarantee for the multilevel estimator to perform better than the single-level estimator here. A weaker inter-level coupling, or a less favourable cost ratio could have led to similar, or even degraded performance. A first set of experiments (not shown) actually yielded a smaller improvement. We tracked it down to the stochastic coupling being too weak for small length-scales, as discussed in Briant et al. (2023). Moving from bilinear to bicubic interpolation operators solved the



issue, as a bicubic interpolation acts as a smoother, removing part of the undesirable fine-scale signal while interpolating to a coarser grid.

Note that the member allocation problem can also be generalized to the problem of selecting a subset of fidelity levels, as using all available levels may not be the optimal solution. In practice, we solved the member allocation problem multiple times, for all possible subsets of levels that include the finest one. The best theoretical variance was obtained with these 4

levels. Adding yet another coarser grid only degraded the accuracy of the estimator.

## 5.2 Empirical variance reduction

This reduction in variance is measured as a global average. To see how this translates into local variance reductions, and to validate the theoretical computations, we explicitly build 200 realizations of a multilevel covariance estimator and apply them to a Dirac impulse. This has the effect of extracting a column of the covariance matrix estimator. Each one of these 200 realizations

uses a different set of coupled ensembles with a total of 603 members per realization (as $603 = 243 + 2 \times 125 + 2 \times 45 + 2 \times 10$, c.f. Fig. 2). From these 200 realizations, we compute an estimate of the statistical expectation and MSE of the estimator, and compare it to an estimate of the bias and MSE of the same-cost 20-member MC estimator. The latter estimates are also based on 200 realizations of the MC covariance estimator, with 20 members per realization.

These estimates are shown in Fig. 4 and Fig. 5. For the expectation of the covariance column, we also show the "true"

expectation, estimated from a $10^4$-member MC estimator. Given that the typical values of stream function fields are of order $10^7$ m$^2$s$^{-1}$, the typical covariances are of order $10^{14}$ m$^4$s$^{-2}$, and their MSEs of order $10^{28}$ m$^8$s$^{-4}$. These large values do not affect the numerical accuracy of the sample estimates shown here, since these estimates were obtained by adding or subtracting values with similar order of magnitude only.

As expected, the empirical expectations all look very similar; no bias is visible in the MC estimator nor in the MLMC one.

The MSE (or equivalently the variance, in this unbiased context) has a smaller amplitude for most grid points when compared to the MC estimator. Some minor degradation can be seen close to the northern and southern boundaries. These degradations are likely due to the forced linear decay of the perturbations within 300km of the North and South boundaries. This can provoke large gradients close to the boundaries, which translate into large values of potential vorticity locally generating high-frequency features. These fine-scale details cannot be accurately captured on the coarse grids, hence the lack of coupling and

bad performance in these areas.

For better visualization, a cross-section of the covariance estimate is shown in Fig. 6. The smaller amplitude of the spread around the true value is obvious for the weighted multilevel estimator. It is consistent with the global factor 1/3 computed in the previous section, as a ratio of 0.337 in variance corresponds to a ratio of $\sqrt{0.337} \approx 0.58$ in standard deviation, which is roughly what is seen in the figure.

## 5.3 From a covariance estimator to B: localization and handling negative eigenvalues

Before using the MLMC covariance estimator in an analysis scheme, we need to address two questions: how to localize it, and how to deal with negative eigenvalues.



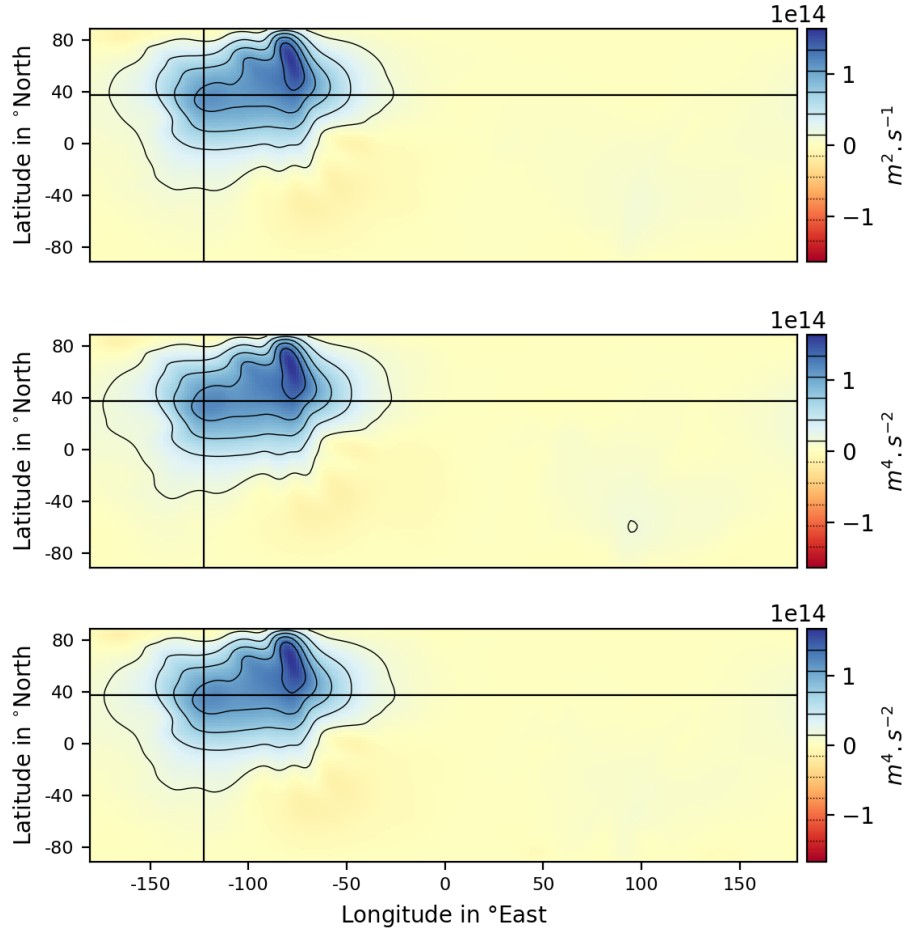

**Figure 4.** Statistical expectation of various estimators of a column of the covariance matrix: True expectation estimated from the 10k MC experiment (top panel), MC estimator with 20 members (middle panel), and wMLMC estimator with same computational budget as a 20-member MC (bottom panel). The horizontal and vertical lines indicate the position of the point with respect to which covariances are computed. Only the bottom model layer is shown.



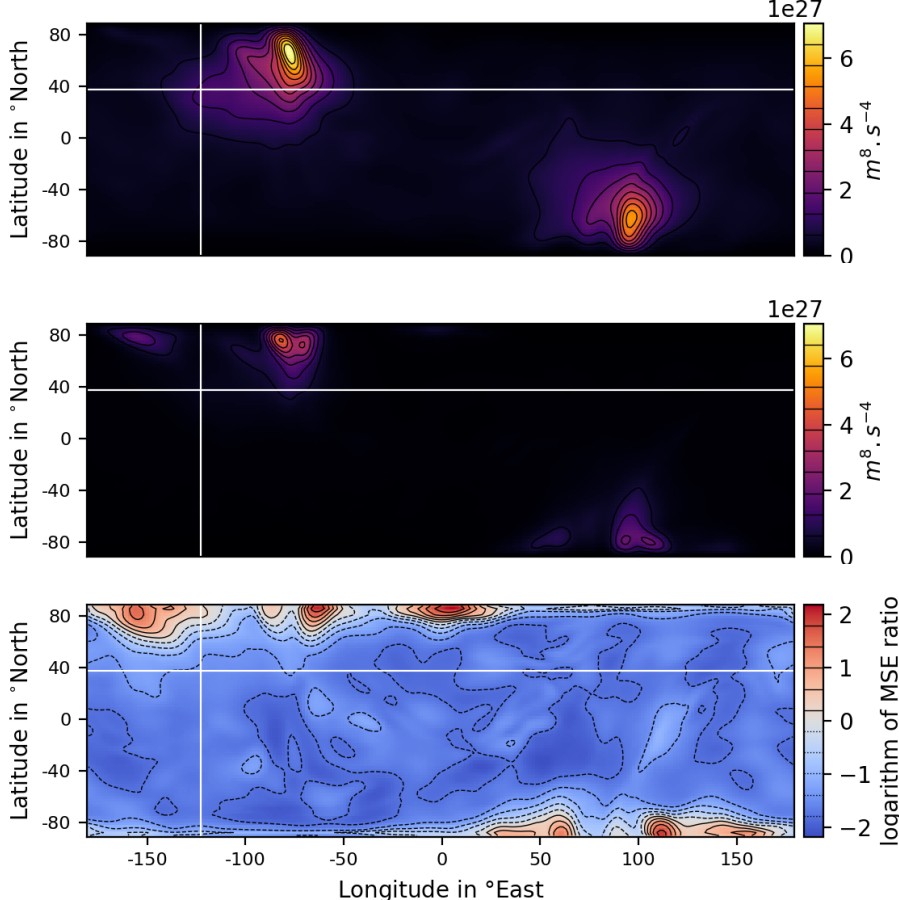

**Figure 5.** Similar to Fig. 4 for the mean-squared error. Top: MC estimator with 20 members; middle: wMLMC estimator with same computational budget; bottom: logarithm of the ratio of MC MSEs over wMLMC MSEs.

Though more accurate, a multilevel ensemble $\mathbf{B}$ estimate is still rank-deficient and still needs some regularization before being used in a variational analysis scheme. To preserve the unbiasedness property of the MLMC $\mathbf{B}$ estimator (compared to the expectation of its localized MC counterpart), we could apply a single common localization to the final covariance estimator. However, in operational ensemble-variational data assimilation, the Schur products are never computed explicitly. To avoid storing the full covariance matrix, it has been shown that covariance operators of the form $\mathbf{X}\mathbf{X}^{\mathrm{T}} \odot \mathbf{L}$ can be computed through a process that only requires storing $\mathbf{X}$ (cf. Appendix B of Buehner, 2005). Since such a decomposition $\mathbf{X}\mathbf{X}^{\mathrm{T}}$ does not exist for the MLMC-estimated $\mathbf{B}$, a localized version of it can only be built by applying localization to each MC estimator in the





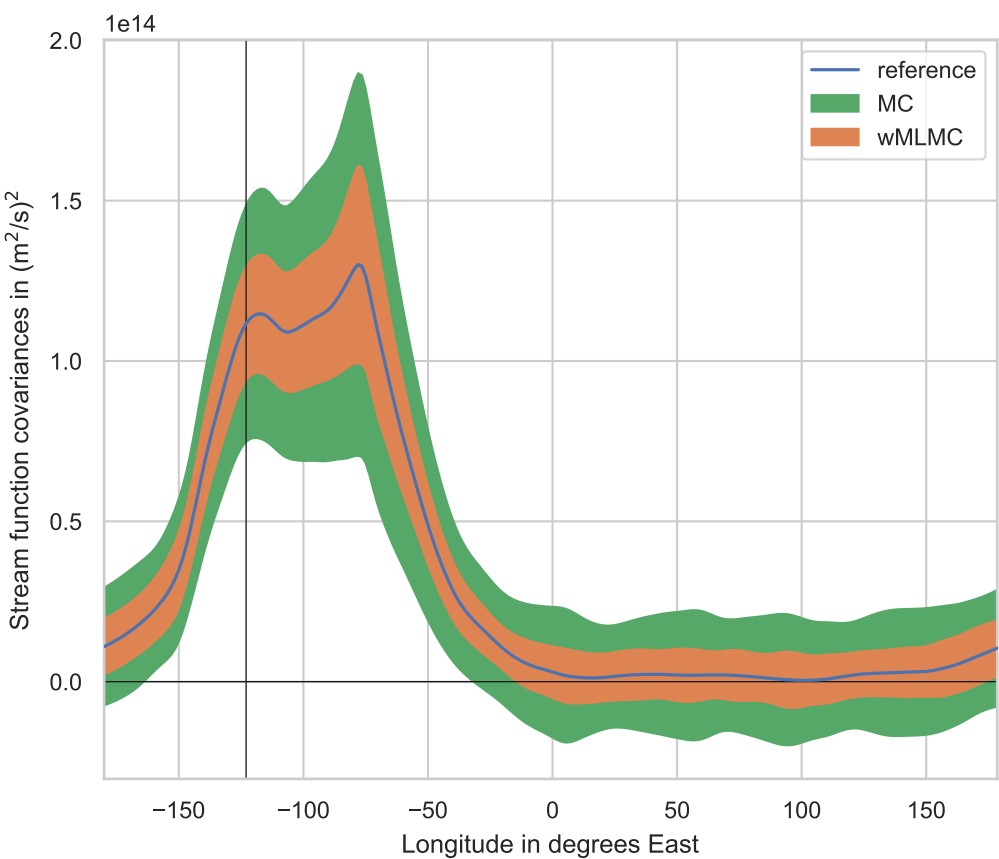

**Figure 6.** Typical spread of the estimators of a column of the covariance matrix. The estimates shown here correspond to a 1D section of Fig. 4 and Fig. 5, at the horizontal latitude line shown on these figures. The shaded area for each estimator gives the $\pm\sigma$ range around the mean, where $\sigma$ is the standard deviation of the estimators.





MLMC telescopic sum:

$$\widehat{\mathbf{B}}_{\mathrm{loc}}^{\mathrm{wML}} = \beta_1^{(1)} \widehat{\mathbf{B}}_1^{(1)} \odot \mathbf{L} + \sum_{k=2}^{L} \left( \beta_k^{(k)} \widehat{\mathbf{B}}_k^{(k)} \odot \mathbf{L} + \beta_{k-1}^{(k)} \widehat{\mathbf{B}}_{k-1}^{(k)} \odot \mathbf{L} \right).$$

As the cost of applying localization to a covariance matrix scales with the ensemble size, localizing some terms like the base term $\widehat{\mathbf{B}}_1^{(1)}$ with hundreds of members could become prohibitively expensive if localization is performed on the fine grid. An alternative is to perform localization on the coarse grid, which is cheaper, and then apply the prolongation operators on the 480 localized increments. Denoting $\widehat{\mathbf{B}}_\ell^{(k)} = \left( \mathbf{P}_{\ell \to L} \mathbf{X}_\ell^{(k)} \right) \left( \mathbf{P}_{\ell \to L} \mathbf{X}_\ell^{(k)} \right)^{\mathrm{T}}$, we have

$$\left( \left( \mathbf{P}_{\ell \to L} \mathbf{X}_\ell^{(k)} \right) \left( \mathbf{P}_{\ell \to L} \mathbf{X}_\ell^{(k)} \right)^{\mathrm{T}} \right) \odot \mathbf{L}_L \approx \mathbf{P}_{\ell \to L} \left( \left( \mathbf{X}_\ell^{(k)} \mathbf{X}_\ell^{(k)^{\mathrm{T}}} \right) \odot \mathbf{L}_\ell \right) \mathbf{P}_{\ell \to L}^{\mathrm{T}},$$

where a subscript has been added to the localization matrices, to indicate on which grid the localization is performed. The two strategies should yield reasonably close results, especially when the localization length-scales are large compared to the largest grid spacings involved. In this article, the small size of the problem under consideration allowed us to perform localization on 485 the fine grid for all these terms, despite the large ensemble sizes involved.

When comparing the costs of MLMC and MC estimators, it should be noted that having comparable ensemble-generation costs does not necessarily guarantee comparable memory-storage requirements, nor comparable costs of applying a (localized) matrix-vector product. Instead of imposing a constraint on the ensemble generation cost in Eq. (16), it would also be possible to impose a constraint on the memory storage requirements, or on the cost of performing a matrix-vector product with localization. 490 If the cost models of these three aspects evolve differently as a function of the fidelity level $\ell$, choosing a different constraint in the member-allocation problem would yield different optimal ensemble sizes, and thus different estimator accuracies.

Since localizing a covariance estimator makes it biased, preserving the unbiasedness of the wMLMC covariance estimator should not be our primary concern when tuning localization for this estimator. This suggests we can possibly use different localization parameters for the different terms of the wMLMC-estimated B, even though the resulting sum would no longer 495 have telescopic expectations (i.e., the terms would no longer average out to zero in expectation). To keep localization tuning feasible, we propose to use two sets of localization parameters only: one for the base term and one for the correction terms. This is based on the fact that the base terms has more members, and so may only need a weaker localization. In addition, the specific nature of correction terms may be better captured by a dedicated localization. The optimal localization parameters chosen for the wMLMC and MC estimators will be described in Sect. 5.4.

Another aspect to consider carefully is the lack of positive semi-definiteness of the (weighted) MLMC covariance estimator. The (weighted) MLMC estimator is not positive definite by construction, due to the presence of negative terms in the correction terms. There is no obvious way to constrain this. As long as negative terms are involved, there will be a chance that some realizations of the estimators have negative eigenvalues. This problem is well-known in multilevel data assimilation, the usual solution being to truncate the spectrum to remove negative eigenvalues, as in Hoel et al. (2016). This is equivalent to 505 projecting the symmetric matrix onto the space of symmetric positive semi-definite matrices, as explained by Higham (2002). More recently, interesting work focused on building a multilevel estimator of a covariance matrix that would be positive by





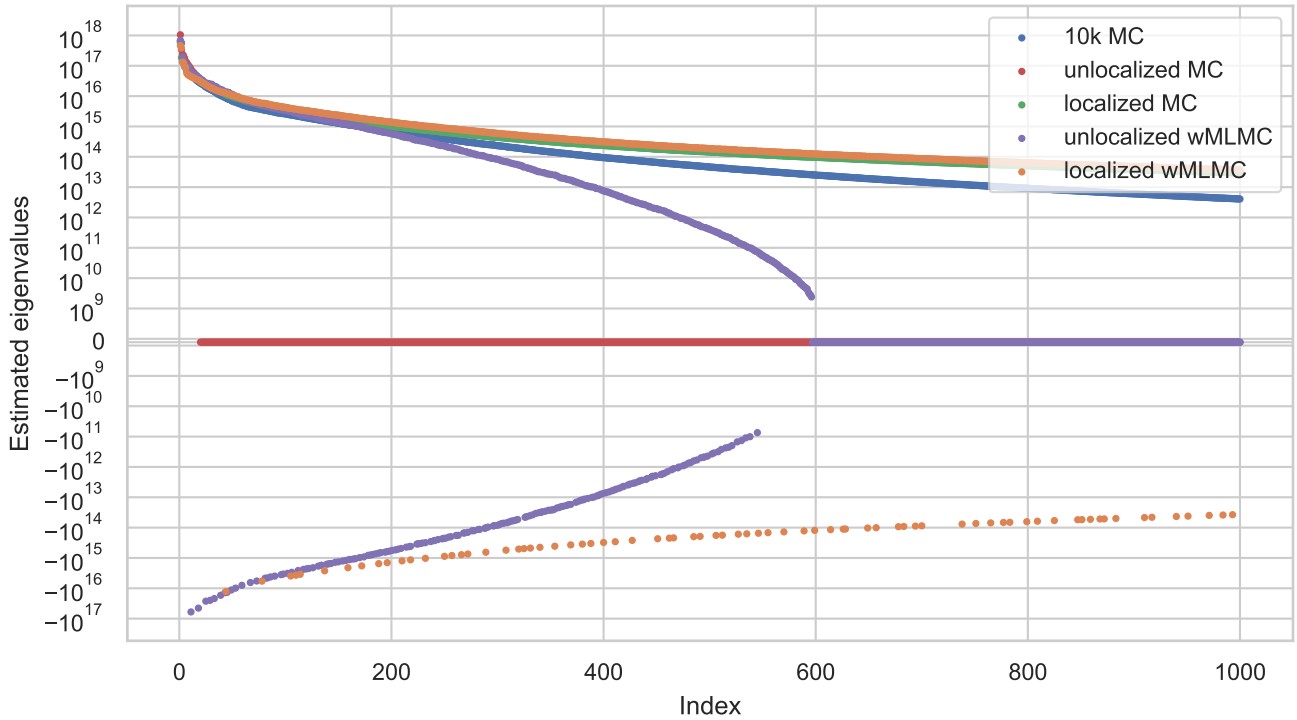

**Figure 7.** Dominant eigenvalues of various covariance matrix estimates. Only the 1000 eigenvalues with largest amplitude (negative or positive) are shown. The y-axis uses a symmetric logarithmic scale, with a linear range from $-10^9$ to $10^9$.

construction (Maurais et al., 2023a, b). As this involves computing costly matrix logarithms and exponentials, the feasibility of this approach for operational data assimilation in high dimensions remains to be demonstrated.

In our case, we do observe negative eigenvalues in wMLMC estimates, as shown in the spectra in Fig. 7. These spectra were obtained for a single wMLMC estimate with and without localization, a standard MC estimate with and without localization, and the reference $\mathbf{B}$ estimated with $10^4$ members (with no localization). The spectra are obtained via a randomized eigenvalue decomposition using 2020 samples, as described in Algorithm 1 of Saibaba et al. (2016).

We first note that the standard MC and wMLMC estimates are both rank-deficient, with ranks 19 and 596 respectively, as evidenced by the zero eigenvalues from indexes 20 and 597. For the unlocalized wMLMC-estimated $\mathbf{B}$, the first negative eigenvalue appears at index 11, with an amplitude of $0.09$ times the first (positive) eigenvalue. Many others follow, to the extent that the full spectrum includes 30% negative eigenvalues. This is clearly not negligible and could cause convergence issues during the minimization of the data assimilation cost function. The situation improves after localization has been applied, with the first negative eigenvalue appearing at index 44 in the spectrum, with amplitude $0.03$ times the first (positive) eigenvalue. The ratio of negative eigenvalues is also decreased from 30% to 7%. Although localization helps, it does not fully solve the problem.



We now explore various possible practical remedies to treat the presence of negative eigenvalues. The first solution, already mentioned, involves truncating the spectrum to remove negative eigenvalues. This implies performing an eigen-decomposition of the covariance matrix and rebuilding it without the eigenvectors associated with negative eigenvalues. This decomposition can be done out of the critical path of a data assimilation suite, before the first observations become available. Based on the

fact that an unlocalized wMLMC-estimated $\mathbf{B}$ has low rank, the numerical cost of the eigen-decomposition could be kept reasonable. For instance, relying on a randomization method, we would only have to apply as many matrix-vector products as the rank of the matrix, and this could be done in parallel. In our case, an upper bound of this rank is known beforehand from the number of members used to build the estimate. For instance, for the wMLMC-estimated $\mathbf{B}$ above, the rank is at most $(N^{(1)} - 1) + 2(N^{(2)} - 1) + 2(N^{(3)} - 1) + 2(N^{(4)} - 1) = 596$, which is small compared to the size of the matrix ($37,920$ here,

c.f. Table 2). However, after removal of negative eigenvalues, there is still a large number of eigenvectors remaining (419 in this case). Contrary to the raw MLMC case, these vectors are now stored on the finest grid, and we are back to the problem of applying localization to a 419-member ensemble on a fine grid. Compared to the original problem of applying localization to a 20-member ensemble, the cost has prohibitively increased. We are thus led to conclude that while randomization approaches may be of interest for offline diagnostics, they are not a viable solution if the cost of applying $\mathbf{B}$ is to remain comparable to the

cost of applying a standard localized ensemble $\mathbf{B}$.

A second solution would be to rely on localization and hybridization. As already mentioned, localization has no theoretical reason to make the matrix positive semi-definite (PSD), although it helps reducing the amplitude and numbers of negative eigenvalues in our case. Hybridization is more promising for this purpose, however. By defining the covariance matrix as a weighted average of an ensemble $\mathbf{B}$ and a parametric $\mathbf{B}$ hybridization can possibly restore positive semi-definiteness in the

covariance estimate. Given the relatively small amplitude of the negative eigenvalues after localization, we can reasonably assume that hybridization could remove most if not all of the negative eigenvalues, even with a small weight for the parametric $\mathbf{B}$. In another context, Higham et al. (2016) proposed an algorithm to estimate the smallest hybrid weight that restores positive semi-definiteness. Whether this is applicable to high-dimensional (but low-rank) problems, and whether this could be done without giving too much weight to the parametric static $\mathbf{B}$ is still to be explored.

A last option, that could be combined with the previous one, would be to use a possibly non-PSD matrix in the minimization algorithms, and adapt the algorithms to deal with these negative eigenvalues. We first remark that using a $\mathbf{B}$-preconditioned conjugate gradient minimization algorithm (Derber and Rosati, 1989) dispenses with the need to define a $\mathbf{U}\mathbf{U}^{\mathrm{T}}$ decomposition of the background error covariance matrix. As a first crude solution to deal with these negative eigenvalues, we propose to use a slightly modified $\mathbf{B}$-preconditioned conjugate gradient algorithm.

This minimization algorithm has originally been designed for use with a PSD $\mathbf{B}$ matrix. Here, we propose to use the non-PSD $\mathbf{B}$ as such, but with two additional early-stopping criteria. At each iteration of the minimization algorithm (Algorithm 2 in Gürol et al., 2014), two intermediate scalar values (inner products) are computed: the curvature along the descent direction and the $\mathbf{B}$-norm of the residual. Both quantities are positive with a PSD $\mathbf{B}$. With a multilevel $\mathbf{B}$ having negative eigenvalues, the "$\mathbf{B}$-norm" is no longer a norm, and either or both quantities can give negative values if the descent direction or the residual

are pointing to directions associated with negative eigenvalues. When this happens, the algorithm is exploring directions of



control space where the estimate of **B** is clearly poor, which justifies early stopping. In practice, these checks can be applied to the numerator and denominator of step 9 in Algorithm 2 of Gürol et al. (2014).

## 5.4   Impact on a single analysis

Although the multilevel estimators computed so far are better than MC estimators in the sense of reducing the MSE measured in the Frobenius norm, there is no guarantee that they would yield better analysis estimates. We now perform various minimization experiments to assess the impact of multilevel background error covariances on the quality of the analysis.

The data assimilation scheme is a 3D ensemble-variational scheme, which means there is no time dimension and that **B** is derived purely from an ensemble (or coupled ensembles), with no parametric (climatological) hybrid component. The observation network consists of direct observations of the stream function at randomly selected grid points. To mimic realistic systems, only 1% of the model grid points are observed. The observation values are derived from the truth run. Perturbations are then added to the observations to simulate observation error. The observation error is Gaussian, with a uniform standard deviation set at the same value as the prior ensemble spread ($\sigma_\mathrm{o} \approx \sigma_\mathrm{b}$), i.e., $\sigma_o = 9 \times 10^6$ m$^2$s$^{-1}$. Note that this value of $\sigma_\mathrm{b}$ has evolved from $6 \times 10^6$ m$^2$s$^{-1}$ at time $t = 0$ h (c.f. Sect. 4.2) to $9 \times 10^6$ m$^2$s$^{-1}$ at $t = 12$ h.

As for previous experiments, three covariance models are compared: a 20-member MC estimator, the same-cost wMLMC with optimal member allocation, and a reference **B** with 10000 members ("10k MC"). Including the reference **B** in these experiments provides a benchmark of the gain that can be achieved by improving the estimation of the background error covariance matrix. In operational situations, we would not know the optimal allocation and optimal weights for wMLMC, as we could not afford to run a pre-processing step to compute them in real-time. Instead, we would have to use a climatological value for the ensemble sizes and weights. The weights could be updated every few cycles, which would partially compensate for the sub-optimality of using climatological ensemble sizes.

Both MC and wMLMC covariance estimators are localized. The localization parameters are tuned on an independent realization of the random observation network, random observation errors and background ensembles to minimize the analysis error. The tuning is performed manually by grid search on the space of localization parameters to minimize the RMSE of the analysis. The chosen localization parameter values are given in Table 3.

**Table 3.** Optimally tuned localization length-scales, expressed in grid points (horizontal localization) or model layers (vertical localization). The length-scales are defined as Daley length-scales (Daley, 1993), which match the standard deviation parameter of the Gaussian-shaped localization functions used here.

| Covariances | horizontal localization | vertical localization |
|:---:|:---:|:---:|
| MC | 25 | 1.7 |
| wMLMC base term | 45 | 1.3 |
| wMLMC correction terms | 30 | 1.3 |
| 10k MC | none | none |



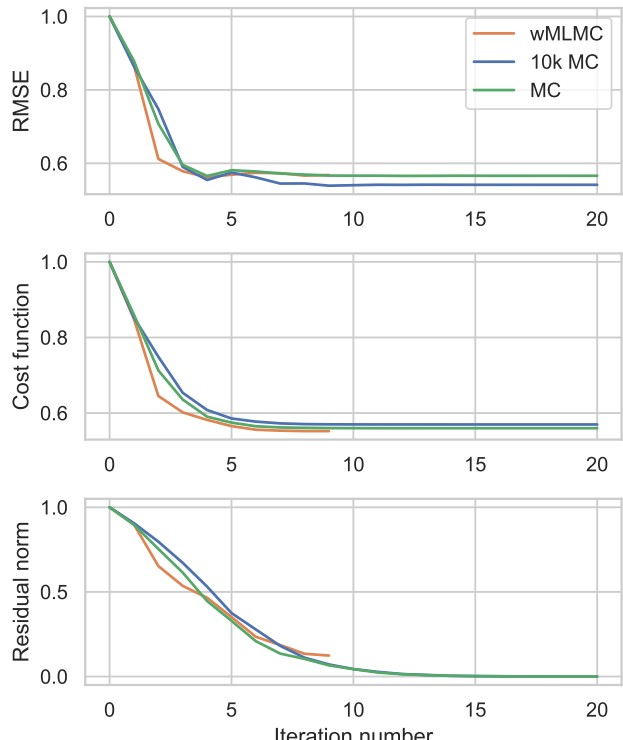

**Figure 8.** Evolution of minimization metrics during the minimization. RMSE with respect to truth (top), cost function (middle) and 2-norm of the residual (bottom). All quantities are normalized by their initial value. Note that in a real application where we do not have access to the truth, only the bottom two metrics are available.

The minimization is performed using the updated **B**-preconditioned conjugate gradient algorithm introduced in the previous

section. Twenty iterations are found to be sufficient to reach convergence in all three settings, as shown in Fig. 8. For this case, a negative "**B**-norm" of the residual led to an early stopping of the wMLMC minimization at iteration 9 out of 20.

Sensitivity to the observation location and errors is removed by running the analyses 200 times with different realizations of the random observation network, random observation errors and random background ensembles (for MC and wMLMC experiments). Whisker plots of the analysis error with respect to the truth run are shown in Fig. 9.

The impact of using wMLMC covariances in the analysis is globally positive for all percentiles shown. However, the magnitude is quite small, with only 1% reduction of analysis error on average (not shown on Fig. 9). If compared to the smallest possible error achieved by the 10k MC experiment for each realization of the observation network, the relative error reduction increases to 2% on average. The median is 10%, suggesting that the low 2% figure is due to a few cases of large deterioration. This 2% improvement is of less importance than the gain observed on the covariance estimator in Sect. 5.1 and 5.2. This is not



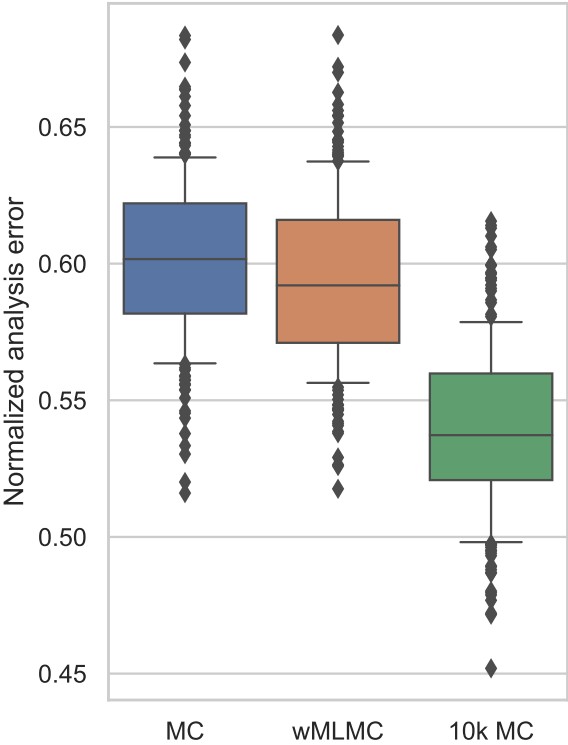

**Figure 9.** Comparison of analysis RMSEs normalized by the RMSE of the background state. The boxes show the quartiles and medians of the 200-realization datasets. The whiskers show the 10th and 90th percentiles.

surprising, as there is no direct relationship between the MSE of the background error covariance estimator and the MSE of the resulting analysis.

## 6  Discussion and conclusions

We have discussed in this article the potential of MLMC methods to improve the estimation of background error covariances for ensemble data assimilation. Starting from an EDA, the main idea is to remove a few ensemble members and to reallocate
the associated computational resources to generate a much larger ensemble on coarser grids, with the goal of obtaining less noisy estimators. By further combining stochastically-coupled ensemble members generated across several grids, multilevel techniques provide estimators that are not only more accurate, but also unbiased.

We have illustrated with a quasi-geostrophic model how optimal ensemble sizes can be determined in a multilevel setting. We have also illustrated how a localized multilevel covariance matrix can be built as a matrix-vector operator. The approach
does not require explicit storing of the covariance matrix elements, nor does it require increasing the number of ensemble





members to be stored at the resolution of the assimilation grid. The proposed method has a computational cost that scales linearly with the state dimension.

However, using the resulting multilevel background error covariance matrix in a variational data assimilation scheme presents certain challenges since, by construction, the matrix does not guarantee positive semi-definiteness. Although various
approaches have been proposed in the literature to enforce PSD multilevel covariance estimates, none of them is computationally affordable for extremely high-dimensional data assimilation problems such as those encountered in NWP. Existing methods either increase the number of ensemble members to be stored on the assimilation grid, or scale at least quadratically with the state size.

To ensure that the method is feasible for high-dimensional problems, we proposed to adapt the minimization algorithm to
account for a non-PSD background error covariance matrix, rather than trying to render the matrix PSD for the minimization algorithm. In a **B**-preconditioned conjugate gradient minimization algorithm, for instance, several diagnostics can be used to detect spurious (negative) eigenvalues in the background error covariance matrix. In our experiments, stopping the minimization early, as soon as negative eigenvalues are encountered, resulted in an improved analysis. This approach offers no guarantee of convergence, however, and further research is needed to make this algorithm more robust. Although not explored
in this study, another promising approach to enforce positive semi-definiteness would be to hybridize the ensemble covariance matrix with a parametric covariance matrix, which is already common in ensemble variational data assimilation.

In addition to the problem of ensuring positive semi-definiteness, several other areas remain unexplored. For instance, it is unclear how the less accurate estimation of small scales in multilevel ensemble covariances would propagate through the analysis and forecast cycles, and how imbalance and spin-up could accumulate or not over cycles. It is also unclear how the
method would perform in a cycled data assimilation/forecast context, where the explicit stochastic inputs would come from perturbations to the innovations or from stochastic physics rather than the initial conditions as considered here.

Finally, it is important to remark that the purpose of an EDA is not only to provide background error covariance estimates to the analysis scheme, but also to initialize an ensemble prediction system (EPS). How a multilevel ensemble data assimilation would impact the subsequent EPS is not known. To minimize the impact on the EPS, one could enforce a minimum ensemble
size on the fine grid. For instance, reducing the number of ensemble members on the fine grid from 20 to 10 may be acceptable if the EPS runs with 10 or fewer members. Another possibility would be to leverage the potential of the multilevel approach for the EPS. In particular, a coupled multilevel EDA could be used to initialize a coupled multilevel EPS. In theory, multilevel techniques could then be used to estimate any statistic of interest: expectations, percentiles, probability of exceeding a threshold etc. This may not be effective in practice, as optimal ensemble sizes are likely to vary depending on the application.
Using a common approach for all applications would imply a weaker coupling for each of these applications, and so reduced improvement in the statistic estimators.

*Code availability.* The code used for the numerical results presented in this article will be made publicly available upon acceptance of the manuscript at [gitlab repository].





## Appendix A: Optimal weights for weighted MLMC

For a given set of ensemble sizes $N^{(k)}$, $1 \leq k \leq L$, Destouches et al. (2023) give a formula for the optimal MLMC weights $\beta_\ell^{(k)*}$, $1 \leq k \leq L$, $\ell = k, k-1$ (see their Sect. 6.2, derived from Schaden and Ullmann, 2020):

$$
\begin{pmatrix} \beta_{k-1}^{(k)*} \\ \beta_k^{(k)*} \end{pmatrix} = \left(\mathbf{C}^{(k)}\right)^{-1} \mathbf{R}^{(k)} \left( \sum_{k=1}^{L} \mathbf{R}^{(k)\mathrm{T}} \left(\mathbf{C}^{(k)}\right)^{-1} \mathbf{R}^{(k)} \right)^{-1} \alpha
$$

where $\mathbf{R}^{(k)}$ is the selection matrix that selects levels $k-1$ and $k$ in a vector of size $L$:

$$
\mathbf{R}^{(k)} \in \mathbb{R}^{2 \times L}, \ R_{i,j}^{(k)} = \begin{cases} 1 & \text{if } i = 1 \text{ and } j = k-1, \\ 1 & \text{if } i = 2 \text{ and } j = k, \\ 0 & \text{otherwise;} \end{cases}
$$

and $\alpha$ is the last vector of the canonical basis of $\mathbb{R}^L$:

$$
\alpha \in \mathbb{R}^L, \ \alpha_i = \begin{cases} 0 & \text{if } i < L, \\ 1 & \text{if } i = L. \end{cases}
$$

*Author contributions.* The contributions of the authors is given following the CRediT taxonomy (https://credit.niso.org/). Conceptualization: P. Mycek (PM) as lead, M. Destouches (MD), S. Gurol (SGu) & A. Weaver (AW) with equal contributions, S. Gratton (SGr) & E. Simon (ES) with supporting contributions. Investigation: MD & PM (equal), SGu, AW, SGr & ES (supporting). Software: MD (lead), SGu (supporting).
Supervision: PM (lead), SGu & AW (equal), SGr & ES (supporting). Writing – original draft: MD. Writing – review & editing: MD, PM, SGu, AW (equal), SGr, ES (supporting).

*Competing interests.* The authors declare that they have no competing interests.

*Acknowledgements.* We are grateful to Benjamin Ménétrier for his help in setting up OOPS-JEDI and the nested grids of the quasi-geostrophic model. We thank Gordon Inverarity, for his informal review of this manuscript. This work was partly supported by the French
national program LEFE (*Les Enveloppes Fluides et l'Environnement*).





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
