# Peer review of "Multilevel Monte Carlo methods for ensemble variational data assimilation"

_EGUsphere, 2024_

## Author Response (AR1)

**Authors' response to Referee #1's comments for "Multilevel Monte Carlo methods for ensemble variational data assimilation "**

Preprint egusphere-2024-3628

M. Destouches, P. Mycek, S. Gürol, A. T. Weaver,
S. Gratton and E. Simon

March 10, 2025

*The authors discussed the background error covariance estimation using (weighted) multi-level Monte Carlo (wMLMC) method in variational data assimilation (DA). The authors discussed several practical considerations when MLMC is sued to estimate a covariance matrix: 1) the mean squared error and variance of a covariance MLMC estimator; 2) computational budget allocation; 3) localisation and positive definiteness of the estimated covariance matrix. The MLMC estimator and the performance of corresponding 3DEnVar is investigated by a two-dimensional two-layer quasi-geostrophic channel model after 12 hour forecast from an initial ensemble without data assimilation cycles. The paper is well-written and is worth publication.*

We thank the referee for their comments and suggestions. In the following, we discuss the major and minor comments, and explain how we have updated or will update the manuscript to address them.

Other modifications, that were not directly asked by the referee, have also been applied to the manuscript. The reason for this is detailed at the end of this document.

*Major comments:*

*1. The Experimental setting section may be benefit from a figure to illustrate the model setup.*

Thank you for this comment, we have added such a figure.

*2. Current results are all built on a single dynamical snapshot of the model. Optionally, is it possible to build a stronger case by running a long deterministic trajectory of the model, and a few select different time step with very different features of dynamics as initial condition to generate ensemble with 12 hour forecast such that the computational cost does not drastically increase?*

The dynamics of the quasi-geostrophic model are not very complex, and the dynamical features observed are very similar across time. As such, we do not expect the conclusions of the paper would be altered by selecting another date to run the experiments.

On another, more practical level, it would indeed be costly to perform the full experiment again on several other situations. There are three type of results in our manuscript: a) the theoretical estimation of the MSE of a $\mathbf{B}$ estimator that can be reached with a multilevel approach (section 5.1), b) the empirical estimation of the MSE reduction that is reached on a column of the $\mathbf{B}$ matrix when actually building these estimators (section 5.2), and c) the impact on a single analysis (section 5.4).

The results of sections 5.2 and 5.4 would be especially cumbersome to reproduce, as they require to run more than 100, 000 forecasts for each date (200 realizations of multilevel estimators using about 600 different forecasts each). However, we can easily reproduce on other dates the experiment of section 5.1 (theoretical MSE reduction on **B**), as it only requires to run 400 forecasts per dates.

We reproduced the results of section 5.1 for 4 other dates, selected 10, 20, 30 and 40 days later than the date studied in our manuscript. The seed of the random number generator was different for each date. The 4 background states are shown in Fig. 1 in this document. The key figures of section 5.1 are reproduced in Fig. 2 and Fig. 3. As summarized in Table1, the variance reductions are of similar order of magnitude than for the case studied in our paper. We do not plan to reproduce all these figures in the paper, but we will mention that the results of this section do not change significantly with the dynamical situation.

[Figure]

Figure 1: Background state at different times, 10 to 40 days later than the background state used in the paper.

| simulation day | MLMLC | wMLMC |
|:---:|:---:|:---:|
| 0 | 63% | 66% |
| 10 | 69% | 72% |
| 20 | 65% | 69% |
| 30 | 66% | 69% |
| 40 | 68% | 71% |

Table 1: Variance reduction achieved by MLMC and weigted MLMC with the same cost as a reference 20-sample MC estimator. First line (day 0) is the result shown in the paper.

*3. When the ensemble member allocation is tuned based on Eq. (14) and (16), do we expect that the $a^{(k)}$ and $b^{(k)}$, or $C^{(k)}$ change significantly due to the flow-dependency of ensemble forecasting?*

[Figure]

Figure 2: Optimal sample allocations associated to the background states in Fig. 1. Corresponds to manuscript Fig. 2.

[Figure]

Figure 3: Theoretical variance of covariance matrix estimators associated to the background states in Fig. 1. Corresponds to manuscript Fig. 3.

*The authors suggest to change the ensemble allocation less frequently (L573 - L575). Do we expect that MSE of estimated B matrix at least as accurate as high-resolution B matrix, i.e. the MC method?*

This is an interesting question. The $a^{(k)}$, $b^{(k)}$ and $C^{(k)}$ coefficients are related to the spatially-averaged variance of the differences between estimators on different fidelity levels. How much they would change with time is not assessed in this manuscript, especially as it would strongly depend on the application. For instance, for a large domain where many small and independent dynamical features are represented, one snapshot of the model may already be representative of a full simulation run. In this context, the space average in these coefficients would imply that they should not vary much with time. Conversely, if the full system can switch from a global attractor to another one, and if the low-fidelity simulators fail to represent correctly one of these attractors, then the inter-level coefficients could change significantly with time. For numerical weather prediction applications for instance, we expect the time-variability of the coefficients to be larger in a limited-area model than in a global model.

If the ensemble member allocation is not updated at every cycle, the allocation becomes sub-optimal, and the MSE of the resulting multilevel estimator becomes larger than the MSE of the *optimal* multilevel estimator. There would be no guarantee in this context that the MSE of the multilevel **B** estimator would be smaller than the MSE of the high-resolution Monte Carlo **B**. More generally, we would like to stress that there is generally no such guarantee even with an optimal allocation. The variance of an optimal multilevel estimator with several fidelity levels can be larger than the variance of the same-cost MC estimator. This could be the case for instance if the coarse-fidelity levels were only weakly correlated to the high-fidelity level, and if their computational costs were only slightly cheaper. This is underlined in the paper as well, in Sect. 3.1.

*4. Using the B matrix from MLMC and MC shows, the results show that the analysis has limited improvements. Could this be related to the smooth streamfunction in the QG model? Would we expect significant differences for other fields, e.g., PV?*

We have not studied this, as the standard way to solve the data assimilation problem for this model is to use streamfunction as the control variable, due to its simpler background error structures and statistics. Performing data assimilation in the potential vorticity space would also be a challenge for positioning errors.

If we performed the experiments using potential vorticity as a control variable, we would expect the performance to be degraded, even on the MSE of the **B** matrix estimator. This can be understood with a scale-separation approach. The large length-scales of the fine-simulation can usually be described accurately by the low-fidelity models, which have a large inter-level correlation in this range of the spectrum. Conversely the coarsest fidelity models are not able to represent fine length-scales, with very small or zero correlations in the high-frequency end of the spectrum. For streamfunction, most of the signal is on the large scales, which ensures a satisfying representations on coarser models. Potential vorticity spectra are much less steep, with much more energy on the fine length-scales. This variable is not suited to a multilevel approach, at least not if the low-fidelity levels are based on coarser space discretizations.

More grounded reasoning around the spectral analysis of multilevel Monte Carlo can be found in the recent preprint by Briant et al. (2023).

*Minor comments:*

*L5: "...affordable We investigate..."* → *"...affordable. We investigate..."*

*L57: "...the Ensemble Kalman Filter(..."* → *"the ensemble Kalman filters ("*

Thanks for spotting these typos. We have corrected them.

*L87: "the composition operator" → "a composition operator"*

We rather modified the sentence into "the function composition operator".

*L159: Perhaps a set should be represented with curly brackets?*

Yes, indeed. This has been modified.

*L160: "...stochastic inputs are all independent..." → "... number of stochastic inputs are all independent..."*

We think the suggested modification would change the meaning of the sentence. We have rephrased this sentence to make it clearer:

> The total $\sum_{k=1}^{L} N^{(k)}$ stochastic inputs are all independent and identically distributed.

has become:

> There are thus $\sum_{k=1}^{L} N^{(k)}$ stochastic inputs in total, all independent and identically distributed.

*L221: The author states that "...related to small fourth-order moments of the correction terms, and so to strong correlations between stochastically-coupled simulations...". Does this mean that adjacent level of model must yield similar outcome? How close should be these levels? Does this also justify the 0 value for level 0?*

Yes, models from adjacent fidelity levels should yield similar outcomes for the multilevel approach to be effective. In practice, in our experiments, the inter-level correlations derived from the space average of inter-level covariances ranged from 0.77 (between the two levels of highest fidelity) to 0.94 (between the two levels of lowest fidelity).

In general, there is no easy rule on how close these fidelity levels should be. One has to go through the process of defining the inter-level correlation coefficients, defining the cost model and finding the optimal member allocation to know what variance reduction can be expected. In the very simple case of the weighted 2-level MLMC estimator of a scalar mean, it can be shown from the MLBLUE formalism (Schaden and Ullmann, 2020) that the variance of the multilevel estimator is:

$$\mathrm{Var}[\widehat{\mu}] = \frac{N^{(\text{low})}(1 - \rho^2) + N^{(\text{high})}}{N^{(\text{high})}\left(N^{(\text{low})} + N^{(\text{high})}\right)}, \tag{1}$$

where $\widehat{\mu}$ is the multilevel estimator with optimal weights, $N^{(\text{low})}$ is the number of samples on the low-fidelity level, $N^{(\text{high})}$ is the number of samples from the high-fidelity level, $\rho$ is the inter-level correlation, and the variance of the random variable is assumed to be 1 without loss of generality. In the limit of infinitely large cost ratio between high and low simulations, i.e., in the limit of infinitely large $N^{(\text{low})}$, we find a variance of $(1 - \rho^2)/N^{(\text{high})}$. Compared to the variance $1/N^{(\text{high})}$ of the same-cost MC estimator, this gives a relative variance reduction of $\rho^2$. So a 0.5 correlation for instance would give 25% variance reduction. Such simple considerations are of little use in practice, given that the cost-ratio is never infinite, more than 2-levels can be used, smaller reduction should be expected for the estimation of larger-order moments that the mean, and uniform scalar weights are suboptimal for a non-scalar problem.

*L270: "corrections term" → "correction term"*

Done.

*L318: "1:3 ratio..." → "1:3 ratio between the width and length of the domain..."*

Done.

*L443: "apply them to a Dirac impulse" reads as if the estimator is used to estimate the covariance of a Dirac impulse, which is not the case, I think. This also means that Fig. 4 and 5 are covariance of one grid point with the entire domain.*

This is what happens, unless I am missing something in the question. Fig. 4 and Fig. 5 are indeed covariances of the entire domain with respect to a single grid point. The general variance reduction estimated in section 5.1 applies to the full covariance matrix estimator, but we can only show the impact on three-dimensional columns of this estimator. Computing the actual impact on a full covariance matrix would have required about $4 \times 10^4$ more memory usage (the number of columns in the covariance matrix), which is prohibitive.

*L477: Here, can the authors briefly explain why the cost is proportial to the ensemble member instead of grid size?*

The cost scales with both the number of ensemble members and the number of grid point in each ensemble member. We will add a citation to a paper explaining why the localization cost scales with the ensemble size (appendix B of Buehner, 2005).

Here, "is to remain" was supposed to convey the idea of a constraint provided by the user and that must be met. The text has been rephrased to make it clearer:

> We are thus led to conclude that while randomization approaches may be of interest for offline diagnostics, they are not a viable solution if the cost of applying $\mathbf{B}$ is to remain comparable to the cost of applying a standard localized ensemble $\mathbf{B}$.

has been rephrased into:

> We are thus led to conclude that while randomization approaches may be of interest for offline diagnostics, they are not a viable solution to the negative eigenvalue problem, unless we allow the cost of applying $\mathbf{B}$ to increase significantly compared to the cost of applying a standard localized ensemble $\mathbf{B}$.

For an SPD matrix $\mathbf{B}$, the $\mathbf{B}$-norm of a vector $\mathbf{u}$ is the norm defined by $\|\mathbf{u}\|^2 = \mathbf{u}^\mathsf{T}\mathbf{B}\mathbf{u}$. The definition is missing in our manuscript, thank you for spotting this. We will correct it.

Yes, indeed. This is now corrected.

**Other modifications**

In addition to the changes suggested by the referee, a few modifications have been applied to the manuscript (visible in the latest version of the manuscript when it will be submitted). When replying to the referee's comments, we realized that Figure 9 had not been obtained with the early-stopping method described in the article. It was a remnant of another solution that we had explored, where we relied on backtracking on the residual norm rather than early-stopping criteria. We had moved away from this backtracking approach as there was little ground for this, but we accidentally kept all figures and data in the paper. Although this has no impact on the conclusions, it did alter some data and figures. This will be corrected in the next submission of the manuscript. We reproduced all the results to be consistent with the early-stopping approach presented in the text:

- There is no clearly visible difference in Figures 1 to 6.

- The conclusions of the paper are not affected.

- The weights of the multilevel estimator given in the paper were not the correct ones, which has been corrected (this is unrelated to the choice of backtracking or early-stopping approach)

- The wMLMC localization parameters tuned for the early-stopping approach are different than those tuned for the backtracking approach.

- As a result, the spectrum of the localized wMLMC covariance matrix estimate in Fig. 7 is different, with less negative eigenvalues than in the previous backtracking approach.

- As the experiments have been fully reproduced in a different computing environment compared to the first submission, the random samples used to build the MLMLC estimator are different. This explains minor differences in the spectra of the unlocalized wMLMC covariance matrix estimate (first negative eigenvalue at index 8 rather than 11, with amplitude 11% rather than 9%, but with no change in the global proportion of negative eigenvalues).

- In Fig. 9, the relative error reduction compared to the best achievable reduction is increased from 2% on average (and 10% on median) to 11% on average (and 13% on median). The message of the figure is not affected.

**References**

Briant, Jérémy et al. (2023). *A filtered multilevel Monte Carlo method for estimating the expectation of discretized random fields*. DOI: 10.48550/arXiv.2311.06069. arXiv: 2311.06069 [math.NA].

Buehner, Mark (2005). "Ensemble-derived stationary and flow-dependent background-error covariances: Evaluation in a quasi-operational NWP setting". In: *Quarterly Journal of the Royal Meteorological Society* 131.607, pp. 1013–1043. DOI: 10.1256/qj.04.15.

Schaden, Daniel and Elisabeth Ullmann (Jan. 2020). "On Multilevel Best Linear Unbiased Estimators". In: *SIAM/ASA Journal on Uncertainty Quantification* 8.2, pp. 601–635. ISSN: 2166-2525. DOI: 10.1137/19M1263534. URL: https://epubs.siam.org/doi/10.1137/19M1263534.

**Authors' response to Alban Farchi's comments for "Multilevel Monte Carlo methods for ensemble variational data assimilation "**

Preprint egusphere-2024-3628

M. Destouches, P. Mycek, S. Gürol, A. T. Weaver,
S. Gratton and E. Simon

March 10, 2025

*In this manuscript, the authors propose to use multilevel Monte Carlo methods to estimate the background error covariance matrix (B) that is used in a variational data assimilation method such as 3D- and 4D-Var. The authors first describe the methodology (how to estimate B using first regular then multi-level Monte Carlo methods) in a pedagogical way. They then show numerical experiments using a low-order model, illustrating the increased accuracy of the estimated B matrix and the resulting increased accuracy of the analysis in a 3D-Var assimilation system.*

*The manuscript is very well written is easy to follow, even though some of the developments can be technical from time to time. I would recommend publication after minor revisions to address a couple remarks.*

We thank Alban Farchi for this detailed review. We especially appreciate all the relevant remarks on how to improve the readability of the figures. We reply below to the questions raised and to the suggestions.

*General comments*

*- Some sub-sections, for example in section 3, are substantial and, even though I do not think that they should be shortened, perhaps they could be split into sub-sub-sections.*

Thank you for this comment. We have split the two longest sub-sections into sub-sub-sections. Section 3.2,"Error reduction and optimal ensemble sizes", has ben split into 3.2.1, "Minimizing an upper bound of the multilevel MSE", and 3.2.2, "Directly minimizing the multilevel MSE". Section 5.3, "From a covariance estimator to **B**: localization and handling negative eigenvalues", has been renamed "From a covariance estimator to **B**" and split in 5.3.1, "Localization of a multilevel ensemble estimator", 5.3.2, "Negative eigenvalues of a multilevel covariance estimator" and 5.3.3, "Handling negative eigenvalues".

*- Small impact in data assimilation experiments (last paragraph of section 5): I have the feeling that this discussion could be extended. Do you think that the conclusions would change in a cycled data assimilation context?*

Our "feeling" is that directly applying this MLMC approach to cycled data assimilation would likely change the results. Most of the unknown lies in how the inter-level correlations between coupled members would evolve across cycles. After many cycles, we expect the system to loose memory of the initial conditions, and the only source of inter-level coupling would be the random perturbations of the innovations and (possibly) of model errors. It is unsure this would be enough to maintain strong inter-level correlations, and is unclear at this stage what the options would be to compensate a possibly weaker coupling.

We reported the core of these comments to the last paragraph of section 5.

*- From figure 1, some numerical artifacts are clearly visible in all grids. You mention that, for l=1 and 2, this is consistent with the fact that the bicubic interpolation scheme does not conserve spatial derivatives. Would this issue be mitigated by using higher order interpolation tools? For l=3 and 4, the numerical artifacts seem to be related to the boundary conditions at the south and north poles. Is this "worrying"?*

Yes, we expect that the issue could be mitigated by using higher-order interpolation tools. Actually, our very first experiments with this quasi-geostrophic toy model were using the default bilinear interpolation, which was yielding even stronger artifacts and smaller inter-level correlations.

You are right to say that some of the numerical artifacts are related to the boundary conditions at the south and north of the domain. Actually, artificial East-West structures can be spotted for $\ell = 1, 2, 3$ at a distance $(\Delta y)_\ell$ of the southern and northern edges. This is related to the way boundary conditions are imposed in this implementation of the model. Large streamfunction structures freely evolve within the domain, with fixed, uniform boundary conditions on the boundaries. This provokes possibly large jumps between the inner domain and the boundaries, both in the fields and in their derivatives. These jumps exacerbate the amplitude of interpolation artifacts at these locations.

We do not think this is worrying, since we do not use directly the potential vorticity fields shown in Figure 1. These artifacts are only visible in potential vorticity fields on the finest grid, after low-to-high interpolation in the streamfunction space. In our experiments, we first interpolate from high to low resolution (in the streamfunction space), run forecasts on coarse grids, and interpolate the streamfunction field back to the high-resolution grid. The analysis is then performed in the space of streamfunction, where these interpolation artifacts are not visible.

Still, this could have been worrying if the multi-fidelity estimator had been built as the average of ensemble estimates from each grid. In this case, systematic interpolation errors could have led to a systematic bias in the covariance estimator, and to a systematic bias in the data assimilation context. With the multilevel Monte Carlo approach, systematic interpolation errors cancel out, and the resulting estimator is guaranteed to have a zero systematic bias, at least before localization.

*Technical remarks and comments*

*- L 122 reference to Eq. (3) → I think you mean the equation (without number) at L 120*

You are right, this has been corrected.

*- L 130 Please define what you mean by a "correlation matrix"*

We have now defined it:

> ...a correlation matrix, i.e., a symmetric positive-definite matrix with unit diagonal,...

*- L 150-152 "We assume that the models are ordered from the least (fidelity level l = 1) to the most accurate (fidelity level l = L), from the computationally cheapest (l = 1) to the most computationally expensive (l = L)." What about the (unlikely) case where model A is more accurate but les computationally expensive than model B?*

If model B is less accurate on all aspects and computationally more expensive than A, it has no value and can be discarded. We can see the point of your remark though. It is not always possible to rank the models, even just in terms of accuracy. A generalization of MLMC, the MLBLUE approach (Multi-Level Best Linear Unbiased Estimator, Schaden and Ullmann, 2020), should be used in this case. The MLBLUE makes no assumptions in the rankings of the models. Given their cost model and the full matrix of covariances between all pairs of levels, it can deduce the structure of the optimal multilevel estimator. For instance, an MLBLUE estimator with three fidelity levels may involve a correction term that is not the difference of two estimators from two adjacent fidelity levels (as in MLMC), but a more general linear combinations of the three estimators from the three fidelity levels. In an educational concern, we have chosen to focus on the MLMC subcase here.

To address this question earlier in the article, we added a sentence at the end of the corresponding paragraph:

> Although this is not the focus of this article, note that there exist multilevel techniques that do not require the fidelity levels to have a natural ordering by accuracy and computational cost (see section 3.3).

*- L 246 Is "natural estimator" the most appropriate term here? What about "standard estimator" or "canonical estimator"?*

We modified to "standard ensemble estimator"

*- L 248-249 "but we have preferred using the biased versions here for the sake of simplicity" what do you mean exactly by "here"? In the numerical experiments of the following sections?*

Yes, we replaced "here" by "in our numerical experiments".

*- L 318-319 "with data points defined at the nodes of the grid" Does it really make a difference if the data points are at the nodes or at the centre of the grid?*

From the point of view of the results, we do not expect this to make a difference. Although it may look like a detail when introducing this high-resolution quasi-geostrophic model, we mention it to ensure the description of the nested grids is clear. The fact that the points are defined at the nodes for all grids ensures the coarse-grid data points are nested within the finer-grid data points. This also makes the presentation of high-to-low interpolations more straightforward, as they become mere selection operators. These interpolations would have been more complex if the data had been defined at cell centers and so if the grids had not been nested.

*- L 324 "centred on a grid point" In American English (which seems to be the convention used in the present manuscript) this should be "centered on a grid point".*

We do not use American English, but British English with the Oxford spelling (-ize suffixes rather than -ise). This is one of the options proposed by the guidelines of the journal (https://www.nonlinear-processes-in-geophysics.net/submission.html#english)

*- L 331-332 The code availability statement is sufficient in my opinion. You could remove these two lines.*

Done.

*- L 336 "Performance is measured from the analysis error with respect to a truth run." The term "analysis error" is a bit vague. I would suggest to use "analysis RMSE" (which is the metric used in Section 5.4). Also, "a truth run" could imply that for a single experiment there are multiple "truths", I would hence suggest to reformulate to "the truth".*

Thanks for these clarifications, we have accepted the suggested changes.

*- L 346-347 "from a Gaussian covariance model" For clarity I would suggest to give the expression for this model.*

Agreed.

*- L 347 "horizontally" and "vertically" This is a bit confusing to me. Naively, I would assume that "horizontally" means in the x-y plane and "vertically" means in the z direction. However here we have only two model levels (ie only two nodes in the z direction), it is hence simpler (better?) to directly give the correlation between both levels and therefore, when reading this sentence I was wondering whether "horizontally" means in the x direction and "vertically" in the y direction. To conclude, in order to avoid potential confusion I would suggest to give the actual correlation between model levels in addition (or perhaps even instead of) to the vertical length scale.*

Your first and "naive" understanding is correct. We agree this could be confusing, and have rephrased it as part of the clarification above. The old formulation:

> This perturbation field is generated from a Gaussian covariance model with length-scales of 1000 km horizontally and 6 km vertically, and with standard deviation $6 \times 10^6$ m$^2$s$^{-1}$. For consistency with the unperturbed North and South boundary conditions, the standard deviations of the covariance model decay linearly to zero within 300 km of the North and South boundaries.

has become:

> This perturbation field is sampled from a normal distribution $\mathcal{N}(\mathbf{0}, \mathbf{A})$ with zero mean and covariance matrix $\mathbf{A}$, which is defined as:

$$\text{For } 1 \leq i, j \leq n, \ A_{ij} = \sigma(y_i)\sigma(y_j) \exp\left(-\frac{(x_i - x_j)^2 + (y_i - y_j)^2}{2d_{\mathrm{h}}^2}\right) \exp\left(-\frac{(z_i - z_j)^2}{2d_z^2}\right) \tag{1}$$

$$\text{For } 0 \leq y \leq L_y, \ \sigma(y) = \sigma \min\left(1, \frac{y}{d_{\mathrm{b}}}, \frac{|L_y - y|}{d_{\mathrm{b}}}\right), \tag{2}$$

> where $(x_i, y_i, z_i)$ are the three-dimensional spatial coordinates of point $i$. This Gaussian spatial covariance model has length-scales of $d_{\mathrm{h}} = 1000$ km horizontally and $d_z = 6$ km vertically, and a standard deviation $\sigma = 6 \times 10^6$ m$^2$s$^{-1}$. In practice, the second exponential decay term in $A_{ij}$, associated to the vertical

length-scale $d_z = 6$ km, is either 1 if $z_i = z_j$ or $\exp\big(5^2/(2 \times 6^2)\big) \approx 0.71$ if $|z_i - z_j| = 5$ km (the distance between the centres of each vertical model layer). For consistency with the unperturbed North and South boundary conditions, the standard deviations of the covariance model decay linearly to zero within $d_b = 300$ km of the North and South boundaries.

*- L 414-416 "we fine-tune the allocation by removing or adding members to ensure we stay below the target budget while having as many ensemble members as possible on each coupling group." How is this "fine-tuning" performed? Using a set of (deterministic) pre-defined rules?*

Yes, this is done automatically using a set of deterministic pre-defined rules. We basically fine-tune the ensemble sizes of each coupling group independently, starting with the most expensive one and ending with the cheapest one. This is slightly more astute than rounding down all sizes to the next integer. The manuscript has been modified to make it clearer that the fine-tuning is not subjective:

> ... we fine-tune the allocation by removing or adding members to ensure we stay below the target budget while having as many ensemble members as possible on each coupling group. This fine-tuning is done automatically, from the most expensive group to the cheapest one.

*- L 492-493 "Since localizing a covariance estimator makes it biased, preserving the un-biasedness of the wMLMC covariance estimator should not be our primary concern" Isn't it impossible to preserve the unbiasedness of the estimator when using localisation? One could understand from the second part of the sentence that it is possible in principle.*

Yes, this part is unclear. We rephrased it from

> Since localizing a covariance estimator makes it biased, preserving the unbiasedness of the wMLMC covariance estimator should not be our primary concern when tuning localization for this estimator. This suggests we can possibly use different localization parameters for the different terms of the wMLMC-estimated B, even though the resulting sum would no longer have telescopic expectations (i.e., the terms would no longer average out to zero in expectation).

to

> Localizing an ensemble covariance estimator makes it biased, be it a MC or a wMLMC estimator. Using the same localization as the MC estimator for all wMLMC terms, as proposed in Eqs. (39) and (40), yields a multilevel estimator with the same bias as the localized MC estimator. Here, we rather choose to use different localization parameters for the different terms of the wMLMC-estimated B, even though the resulting sum no longer has telescopic expectations (i.e., the terms no longer average out to zero in expectation).

*- L 524 "out of the critical path of a data assimilation suite" I would suggest to define what you mean here by "critical path".*

Agreed. Here is the modified text:

> This decomposition can be done before the critical path of an operational data assimilation suite, i.e. before the operationally time-constrained interval between the reception of the observations to assimilate and the delivery of the associated analysis to the subsequent forecast.

*- L 539 "...parametric B hybridization..." -¿ "parametric B, hybridization"*

Done.

*- L 642 "The contributions of the authors is given..." -¿ "The contributions of the authors are given..."*

Done.

*- Variances are sometimes written "\mathcal{V}" and sometimes "Var". Please be consistent.*

Yes indeed, sorry about this. This has been corrected.

*- I strongly advise to give a number to all equations. Indeed, even if a specific equation is not very important and not mentioned in the present manuscript, readers (or authors of potential follow-up papers) may need to refer to it. Furthermore, every equation should end with a punctuation sign (",", ".", ";", etc.)*

Thanks for the suggestion, which we accept.

*- Table 1: I would suggest to use the "booktabs" package, with no vertical separation between columns, but with top and bottom horizontal lines and with potential horizontal separators (eg here between row 1 and row 2).*

We now use the "booktabs" package, although we cannot guarantee this will be kept in the editor's version, as it is not included in the default packages proposed by the Copernicus template. We modified the cell separations as suggested by the referee.

*- Table 2: same remarks as table 1. In addition, I would suggest to use right alignment for numbers (which makes it easier to compare rows). In the fourth column the time unit (min) could be included in the column header. For the last column, I would suggest to use the same number of digits per row (again, to make it easier to compare rows).*

Done

*- Figure 1: a color bar is missing. The y labels could use the explicit value of l. The caption could mention "epsilon_1" and "epsilon_2" as the two ensemble members.*

Done

*- Figure 2: "forecast cost" is called "normalized integration cost" previously, right?*

Yes, you are right. We modified the title to be consistent with the text.

*- Figure 4: I would suggest to add a title to each subplot to indicate which estimate is plotted. Also, is the colormap properly centered on 0?*

The titles have been added. Yes, the colormap was properly centered, with yellow as a central colour. We have changed the colormap to a more standard diverging colormap.

*- Figure 5: Same remarks as for figure 4. In addition, I would suggest to reverse the colormap for the top two panels (this would be better when printing the page).*

Done.

*- Figure 6: The caption mention "shaded area", but the figure uses filled areas.*

We corrected the caption.

*- Figure 7: I would suggest to use 2 or 3 columns for the legend (to avoid hiding some part of the figure). The triple horizontal line at y=0 is a bit weird. I would also suggest to increase the size of the markers in the legend (as such, it is very difficult to distinguish between blue and purple).*

Done.

*- Figure 8: Whisker plots can be sometimes a bit deceiving. You could potentially use violin plots instead (also to be fair violin plots have their own limitations as well).*

We had initially considered violin plots, but moved to box plots as violin plots have internal smoothing parameters whose choice can be hard to justify. We modified the figure to have violin plots **and** whisker boxes within the violin plots. We have kept the default values for the violin plot smoothing.

**References**

Schaden, Daniel and Elisabeth Ullmann (Jan. 2020). "On Multilevel Best Linear Unbiased Estimators". In: *SIAM/ASA Journal on Uncertainty Quantification* 8.2, pp. 601–635. ISSN: 2166-2525. DOI: 10.1137/19M1263534. URL: https://epubs.siam.org/doi/10.1137/19M1263534.